# Better Together: Leveraging Unpaired Multimodal Data for Stronger Unimodal Models

**Sharut Gupta**[†], **Shobhita Sundaram**[†], **Chenyu Wang**[†], **Stefanie Jegelka**[† ‡], **Phillip Isola**[†]

[†]MIT CSAIL, [‡]TU Munich

{sharut, shobhita, wangchy, stefje, phillipi}@mit.edu

**Editors:** Marco Fumero, Clementine Domine, Zorah Lähner, Irene Cannistraci, Bo Zhao, Alex Williams

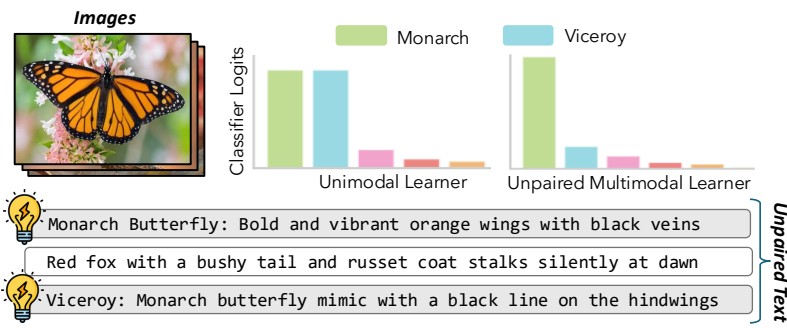

Figure 1: Text provides complementary information beyond images, even when not paired directly; We introduce *Unpaired multimodal representation learning*, a framework that leverages unpaired multimodal data to improve unimodal representations

## Abstract

Traditional multimodal frameworks emphasize learning unified representations for tasks such as visual question answering, typically requiring paired, aligned data. However, an overlooked yet powerful question remains: can one leverage auxiliary *unpaired* multimodal data to directly enhance representation learning in an *individual* modality? To explore this, we propose UML: UNPAIRED MULTIMODAL LEARNER, a modality-agnostic training paradigm in which a single model alternately processes inputs from different modalities—including images, text, audio, or video—while sharing model weights across these modalities. Our approach exploits shared structure in unaligned multimodal signals, eliminating the need for paired data. We show that unpaired text improves image classification, and that other auxiliary modalities likewise enhance both image and audio tasks.

## 1 Introduction

The pursuit of a succinct representation of reality across text, images, and audio has long guided efforts in building intelligent agents. Recent web-scale multimodal learning aligns diverse modalities in a shared latent space [Radford et al., 2021, Singh et al., 2022, Mizrahi et al., 2023, Girdhar et al., 2023a, Bachmann et al., 2022, Li et al., 2023, Bachmann et al., 2024, Jia et al., 2021], capturing cross-modal structure and surpassing unimodal baselines in zero-shot transfer and cross-modal retrieval.

Proceedings of the III edition of the Workshop on Unifying Representations in Neural Models (UniReps 2025).

Yet most approaches overwhelmingly rely on massive paired corpora (e.g., image–caption data) to learn such representations, limiting scalability in specialized or low-resource domains, where paired data is costly. While paired data is expensive to collect and curate, unpaired data is naturally abundant. This raises a critical question:

*Can we move beyond the rigid paradigm of paired learning and meaningfully enhance unimodal representations by accessing unpaired data from other modalities?*

Recent work posits a shared statistical model of reality—an empirical echo of Plato's ideal Forms—where embeddings across modalities converge toward a unified representation as networks scale [Huh et al., 2024, Huang et al., 2021]. Thus, with paired supervision, models can exploit natural co-occurrence to capture shared semantics more accurately. Crucially, however, achieving this convergence does not necessarily require explicit pairs; if each modality samples the same underlying latent space, aligning their marginal distributions can reveal the common semantic structure [Timilsina et al., 2024, Sturma et al., 2023].

Building on this insight, we introduce *Unpaired Multimodal Representation Learning*, a framework leveraging unpaired multimodal data to improve unimodal representations. We first show that theoretically, under linear assumptions, this yields more informative representations than single-modality learning. In some cases, adding complementary modalities yields more accurate ground-truth estimates than merely enlarging the primary dataset—for example, an image classifier may benefit more per sample from training on sentences than on additional images, even when evaluated on images. We instantiate this idea with UML: UNPAIRED MULTIMODAL LEARNER, a shared-weight network that processes diverse modalities with shared weights, distilling synergistic knowledge without explicit pairings. We start with state-of-the-art image and text encoders (e.g., DINOv2, OpenLLaMa) and train a shared network that treats projected data from the second modality as extra training samples, yielding substantial gains on 10 image and audio benchmarks. We further show effective knowledge transfer by initializing vision models with pretrained language model weights. Finally, we quantify an "exchange rate" between images and sentences, measuring how many sentences equal one image for optimal performance. Together, these results show that fully unpaired data from one modality can strengthen another, enabling robust multimodal training without paired supervision.

To summarize, the contributions of our work are:

- We introduce UML: a modality-agnostic shared-weight framework that leverages unpaired auxiliary modalities as extra training data, enriching unimodal features and delivering strong gains across 11 image and audio benchmarks with pretrained encoders (e.g., DINOv2, AudioCLIP). We also quantify conversion ratios between images and sentences, mapping how data from one modality substitutes for another in training.

- Theoretically, under a linear data-generating model, we prove that unpaired auxiliary modalities can strictly reduce estimator variance, yielding more informative representations than any single-modality training.

## 2   Unpaired Multimodal Representation Learning

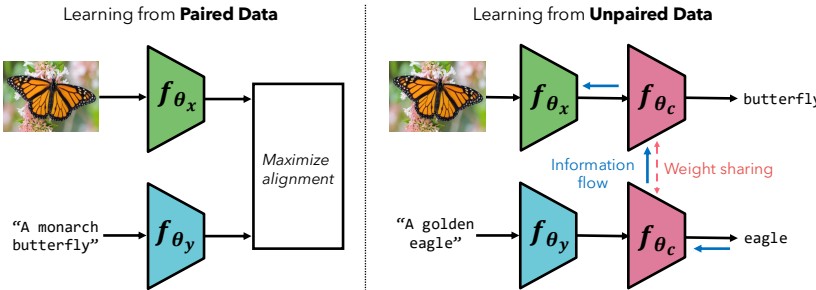

Figure 2: (Left) Traditional paired multimodal learn by maximizing the alignment of paired data across modalities; (Right) We propose UML, which learns from unpaired multimodal data via weight sharing, allowing information flow across modalities to enrich unimodal representations.

We assume an existence of an underlying reality $\mathcal{Z}^*$ observed through different projections (modalities) such as images, text, or audio [Huh et al., 2024, Timilsina et al., 2024, Sturma et al., 2023]. Each modality is represented by a random variable, e.g., $\mathcal{X}$ for images and $\mathcal{Y}$ for text, producing observations $x \in \mathcal{X}$ and $y \in \mathcal{Y}$.

*Multimodal representation learning* seeks encoders $f_{\theta_x} : \mathcal{X} \to \mathcal{Z}$ and $f_{\theta_y} : \mathcal{Y} \to \mathcal{Z}$ mapping each modality into a shared embedding space $\mathcal{Z}$. In the standard setting, training relies on *paired data*, i.e., samples $(x, y)$ drawn from the joint distribution $P(\mathcal{X}, \mathcal{Y})$, which are assumed to be aligned views of the same underlying entity in $\mathcal{Z}^*$ [Radford et al., 2021, Zhai et al., 2023, Jia et al., 2021, Singh et al., 2022, Li et al., 2019, Lu et al., 2019, Bachmann et al., 2022, Roy et al., 2025, Wang et al., 2024, Liang et al., 2023].

In contrast, here we study the setting where such pairs are absent. Crucially, observations $x \sim P(\mathcal{X})$ and $y \sim P(\mathcal{Y})$ are drawn independently, with no guarantee of correspondence. Our objective is still to learn $f_{\theta_x}$ and $f_{\theta_y}$ into a common space $\mathcal{Z}$ that captures the shared structure across modalities. We refer to this setting as *Unpaired Multimodal Representation Learning*. The key insight is that even when unpaired, modalities could carry complementary signals about the same underlying reality $\mathcal{Z}^*$. For example, shuffling text captions across an image dataset destroys pairwise correspondences but still preserves distributional cues that can guide representation learning. We formalize this framework under linear assumptions in Appendix C. Here, estimating the underlying reality is governed by the Fisher information matrix, which measures how sharply the likelihood "curves" around the true $\theta$. Because the two modalities are independent conditioned on $\mathcal{Z}$, their curvature contributions add pointwise, resulting in the joint Fisher information being simply the sum of the unimodal blocks. Thus, loosely speaking, any nonzero contribution from the unpaired $\mathcal{Y}$-samples strictly increases curvature—and thus strictly tightens the variance bound—along those directions. Building on this, we show in Appendix E.10 that the effect can be so pronounced that a single $\mathcal{Y}$ sample may contribute more information than an additional $\mathcal{X}$ sample—i.e., the effective exchange rate between modalities can be below one, even when evaluation is on $\mathcal{X}$.

## 3   UML: UNPAIRED MULTIMODAL LEARNER

Suppose we train on a primary modality $\mathcal{X}$ while leveraging unpaired data from an auxiliary modality $\mathcal{Y}$. We assume access to pretrained encoders $f_{\theta_x}$ and $f_{\theta_y}$ that capture the semantic structure of each modality. Our algorithm maps embeddings from both modalities into a shared latent space and trains a *single* head $h_\theta$ across them. Concretely, given batches $b_x \sim P(\mathcal{X})$ and $b_y \sim P(\mathcal{Y})$, we compute embeddings $z_x = f_{\theta_x}(b_x)$ and $z_y = f_{\theta_y}(b_y)$. These are then passed through the shared head to produce $h_\theta(z_x)$ and $h_\theta(z_y)$, each supervised with its own task-specific loss. For classification tasks, this specializes to

$$\mathcal{L} = \ell_{\text{CE}}(h_\theta(z_x), c_x) + \ell_{\text{CE}}(h_\theta(z_y), c_y),$$

where $c_x$ and $c_y$ are the corresponding class labels for $b_x$ (say images) and $b_y$ (say texts) respectively.

Although supervision is modality-specific, the shared head $h_\theta$ receives updates from both modalities. Consequently, gradients from $h_\theta$ also flow into $f_{\theta_x}$, effectively transferring information from $f_{\theta_y}$. To distill knowledge from text into vision, for example, we freeze the text encoder $f_{\theta_y}$ and use it as a stable semantic source. This shared-weight design allows knowledge from $\mathcal{Y}$ to regularize and enrich representations of $\mathcal{X}$, even without paired samples. Pseudocode is provided in Appendix D.

## 4   Experimental Results and Discussion

We evaluate UML on visual classification in two settings: (1) Full fine-tuning: initializing from a pretrained vision backbone and updating all parameters on the target dataset. (2) Few-shot linear probing: freezing the vision backbone and training a linear classifier on $k$ labeled samples per class ($k = 4$). In both cases, we enrich image representations with unpaired text embeddings, using DINOv2 as the vision encoder and OpenLLaMA as the text encoder. To construct conceptually related yet unpaired text data, we generate text templates with varying amounts of semantic information about the dataset. For further details and specific prompts, refer to Appendix B.3. Our method has two variants: Ours (UML) where we alternately train with both image and unpaired text data (see **??**) and Ours (init) where we initialize the linear classifier with the average text embedding of each class, providing a strong prior to align image and class level information.

| Shot | Method | Cars | FGVC | DTD | UCF101 |
|------|--------|------|------|-----|--------|
| N/A | Unimodal | 79.45 | 66.99 | 72.16 | 83.18 |
| | Ours | 84.87 ↑ | 71.54 ↑ | 74.14 ↑ | 84.77 ↑ |
| | Ours (init) | 86.39 ↑ | 73.44 ↑ | 74.27 ↑ | 84.69 ↑ |
| 4 | Unimodal | 38.76 | 32.10 | 59.69 | 67.75 |
| | Ours | 41.69 ↑ | 33.38 ↑ | 61.58 ↑ | 69.60 ↑ |
| | Ours (init) | 43.17 ↑ | 33.86 ↑ | 62.43 ↑ | 71.13 ↑ |

Table 1: Full finetuning (Shot = N/A) and 4-shot linear probing with ViT-S/14 DINOv2 and OpenLLaMA-3B.

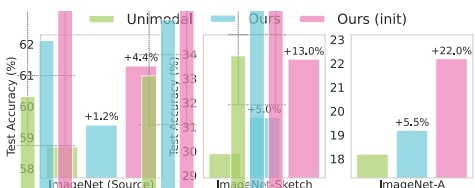

Figure 3: Robustness under test-time distribution shifts. Our approach is more robust than its unimodal counterpart across distribution-shifted test sets.

**Unpaired Textual Data Improves Visual Classification.** As shown in Table 1, across both full fine-tuning and 4-shot linear probing, UML consistently improves over unimodal baselines on Stanford Cars, FGVC Aircraft, DTD, and UCF101. Results on other shots, datasets, model scales, and prompt variants are reported in Appendix E. We also evaluate the robustness of UML-trained models under test-time distribution shifts. A 4-shot linear probe with DINOv2 is trained on ImageNet and tested on ImageNet-Sketch and ImageNet-A. UML consistently outperforms the unimodal baseline (Figure 3), showing that language priors yield more transferable features. Additional robustness results are provided in Appendix E.3.

**Unpaired Image and Text Data Improves Audio Classification.** We extend this analysis beyond image and text modalities to ImageNet-ESC-27 benchmark [Lin et al., 2023]. As shown in Figure 4, UML leverages unpaired image and text data to consistently improve 4-shot audio classification over unimodal baselines. Symmetrically, audio and text also improve image classification, with detailed results, including other shot counts, dataset variants, and model settings, reported in Appendix E.9.

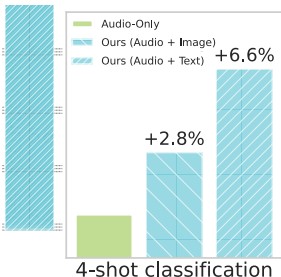

Figure 4: UML improves audio classification using unpaired image and text samples

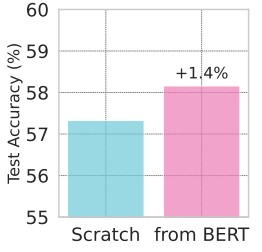

Figure 5: Image classifier trained from BERT initialization outperforms training from scratch

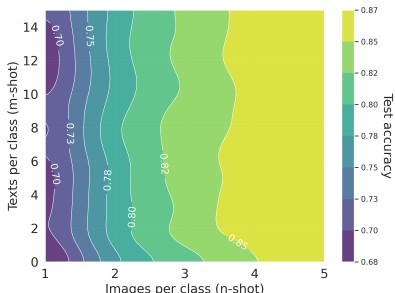

Figure 6: 1 img ≈ 1034 words (DI-NOv2)

**Transfer Learning.** Thus far, we have explored how co-training with multiple unpaired modalities improves the learned representation. Here, we study if *transferring* knowledge from one modality can enhance performance in another by initializing a ViT [Dosovitskiy et al., 2020] with pretrained BERT [Devlin et al., 2019] weights and evaluating on ImageNet (details in Appendix B.4.3). As shown in Figure 5, initializing with BERT weights boosts performance for both frozen and unfrozen backbones. Our results indicate that the semantic knowledge of language models provides a strong initialization for vision.

**Marginal Rate-of-Substitution Between Modalities.** How many words is an image worth? Figure 6 illustrates this by plotting test accuracy isolines on Oxford-Pets (few-shot linear probe), effectively mapping the number of text shots equivalent to a single image shot. Aligned CLIP (1 image ≈ 228 words) as shown in Figure 27 is more efficient than non-aligned DINOv2 (1 image ≈ 1034 words). Indeed, in some cases, an image may quite literally be worth a thousand words. However, we do not control for increasing complexity, so adding sentences does not guarantee extra information. Extended results on aligned CLIP and additional datasets, are provided in Appendix E.4.

## 5 Conclusions and Limitations

In this work, we introduced *Unpaired Multimodal Representation Learning*, showing theoretically and empirically that unpaired data from auxiliary modalities strengthens unimodal representations and enables learning a more accurate representation of the underlying reality. Empirically, we show performance gains across vision and audio classification benchmarks using auxiliary modalities and estimate useful conversion ratios between them. Our algorithm provides a new perspective on how to harness the abundance of unpaired data to learn better representations. While our study focuses on classification, extending to self-supervised objectives (e.g., masked prediction) and text tasks offers promising directions for future work.

## 6 Acknowledgments

This research was sponsored by the Department of the Air Force Artificial Intelligence Accelerator under Cooperative Agreement Number FA8750-19-2-1000, in part by the NSF AI Institute TILOS (NSF CCF-2112665) and the Alexander von Humboldt Foundation. This work was also supported by a Packard Fellowship to P.I., and by ONR MURI grant N00014-22-1-2740. S.G. is supported by the MathWorks Engineering Fellowship. S.S. is supported by an NSF GRFP fellowship. The views and conclusions contained in this document are those of the authors and should not be interpreted as representing the official policies, either expressed or implied, of the Department of the Air Force or the U.S. Government. The U.S. Government is authorized to reproduce and distribute reprints for Government purposes, notwithstanding any copyright notation herein.

# References

Amjad Almahairi, Sai Rajeshwar, Alessandro Sordoni, Philip Bachman, and Aaron Courville. Augmented cyclegan: Learning many-to-many mappings from unpaired data. In *International conference on machine learning*, pages 195–204. PMLR, 2018.

Roman Bachmann, David Mizrahi, Andrei Atanov, and Amir Zamir. Multimae: Multi-modal multitask masked autoencoders. In *European Conference on Computer Vision*, pages 348–367. Springer, 2022.

Roman Bachmann, Oğuzhan F Kar, David Mizrahi, Ali Garjani, Mingfei Gao, David Griffiths, Jiaming Hu, Afshin Dehghan, and Amir Zamir. 4m-21: An any-to-any vision model for tens of tasks and modalities. *Advances in Neural Information Processing Systems*, 37:61872–61911, 2024.

Lukas Bossard, Matthieu Guillaumin, and Luc Van Gool. Food-101–mining discriminative components with random forests. In *Computer vision–ECCV 2014: 13th European conference, zurich, Switzerland, September 6-12, 2014, proceedings, part VI 13*, pages 446–461. Springer, 2014.

Rakesh Chada, Zhaoheng Zheng, and Pradeep Natarajan. Momo: A shared encoder model for text, image and multi-modal representations. *arXiv preprint arXiv:2304.05523*, 2023.

Mircea Cimpoi, Subhransu Maji, Iasonas Kokkinos, Sammy Mohamed, and Andrea Vedaldi. Describing textures in the wild. In *Proceedings of the IEEE conference on computer vision and pattern recognition*, pages 3606–3613, 2014.

Pinar Demetci, Rebecca Santorella, Björn Sandstede, William Stafford Noble, and Ritambhara Singh. Scot: Single-cell multi-omics alignment with optimal transport. *Journal of Computational Biology*, 29(1):3–18, 2022. doi: 10.1089/cmb.2021.0446. URL https://doi.org/10.1089/cmb.2021.0446. PMID: 35050714.

Jia Deng, Wei Dong, Richard Socher, Li-Jia Li, Kai Li, and Li Fei-Fei. Imagenet: A large-scale hierarchical image database. In *2009 IEEE conference on computer vision and pattern recognition*, pages 248–255. Ieee, 2009.

Jacob Devlin, Ming-Wei Chang, Kenton Lee, and Kristina Toutanova. Bert: Pre-training of deep bidirectional transformers for language understanding. In *Proceedings of the 2019 conference of the North American chapter of the association for computational linguistics: human language technologies, volume 1 (long and short papers)*, pages 4171–4186, 2019.

Alexey Dosovitskiy, Lucas Beyer, Alexander Kolesnikov, Dirk Weissenborn, Xiaohua Zhai, Thomas Unterthiner, Mostafa Dehghani, Matthias Minderer, Georg Heigold, Sylvain Gelly, et al. An image is worth 16x16 words: Transformers for image recognition at scale. *arXiv preprint arXiv:2010.11929*, 2020.

Li Fei-Fei, Rob Fergus, and Pietro Perona. Learning generative visual models from few training examples: An incremental bayesian approach tested on 101 object categories. In *2004 conference on computer vision and pattern recognition workshop*, pages 178–178. IEEE, 2004.

Peng Gao, Shijie Geng, Renrui Zhang, Teli Ma, Rongyao Fang, Yongfeng Zhang, Hongsheng Li, and Yu Qiao. Clip-adapter: Better vision-language models with feature adapters. *International Journal of Computer Vision*, 132(2):581–595, 2024.

Xinyang Geng, Hao Liu, Lisa Lee, Dale Schuurmans, Sergey Levine, and Pieter Abbeel. Multimodal masked autoencoders learn transferable representations. *arXiv preprint arXiv:2205.14204*, 2022.

Rohit Girdhar, Mannat Singh, Nikhila Ravi, Laurens Van Der Maaten, Armand Joulin, and Ishan Misra. Omnivore: A single model for many visual modalities. In *Proceedings of the IEEE/CVF conference on computer vision and pattern recognition*, pages 16102–16112, 2022.

Rohit Girdhar, Alaaeldin El-Nouby, Zhuang Liu, Mannat Singh, Kalyan Vasudev Alwala, Armand Joulin, and Ishan Misra. Imagebind: One embedding space to bind them all. In *Proceedings of the IEEE/CVF conference on computer vision and pattern recognition*, pages 15180–15190, 2023a.

Rohit Girdhar, Alaaeldin El-Nouby, Mannat Singh, Kalyan Vasudev Alwala, Armand Joulin, and Ishan Misra. Omnimae: Single model masked pretraining on images and videos. In *Proceedings of the IEEE/CVF conference on computer vision and pattern recognition*, pages 10406–10417, 2023b.

Andrey Guzhov, Federico Raue, Jörn Hees, and Andreas Dengel. Esresne (x) t-fbsp: Learning robust time-frequency transformation of audio. In *2021 International Joint Conference on Neural Networks (IJCNN)*, pages 1–8. IEEE, 2021.

Yu Huang, Chenzhuang Du, Zihui Xue, Xuanyao Chen, Hang Zhao, and Longbo Huang. What makes multi-modal learning better than single (provably). *Advances in Neural Information Processing Systems*, 34:10944–10956, 2021.

Minyoung Huh, Brian Cheung, Tongzhou Wang, and Phillip Isola. The platonic representation hypothesis. *arXiv preprint arXiv:2405.07987*, 2024.

Chao Jia, Yinfei Yang, Ye Xia, Yi-Ting Chen, Zarana Parekh, Hieu Pham, Quoc Le, Yun-Hsuan Sung, Zhen Li, and Tom Duerig. Scaling up visual and vision-language representation learning with noisy text supervision. In *International conference on machine learning*, pages 4904–4916. PMLR, 2021.

Cijo Jose, Théo Moutakanni, Dahyun Kang, Federico Baldassarre, Timothée Darcet, Hu Xu, Daniel Li, Marc Szafraniec, Michaël Ramamonjisoa, Maxime Oquab, et al. Dinov2 meets text: A unified framework for image-and pixel-level vision-language alignment. *arXiv preprint arXiv:2412.16334*, 2024.

Jing Yu Koh, Ruslan Salakhutdinov, and Daniel Fried. Grounding language models to images for multimodal inputs and outputs. In *International Conference on Machine Learning*, pages 17283–17300. PMLR, 2023.

Jonathan Krause, Michael Stark, Jia Deng, and Li Fei-Fei. 3d object representations for fine-grained categorization. In *Proceedings of the IEEE international conference on computer vision workshops*, pages 554–561, 2013.

Guillaume Lample, Myle Ott, Alexis Conneau, Ludovic Denoyer, and Marc'Aurelio Ranzato. Phrase-based & neural unsupervised machine translation. *arXiv preprint arXiv:1804.07755*, 2018.

Liunian Harold Li, Mark Yatskar, Da Yin, Cho-Jui Hsieh, and Kai-Wei Chang. Visualbert: A simple and performant baseline for vision and language. *arXiv preprint arXiv:1908.03557*, 2019.

Yanghao Li, Haoqi Fan, Ronghang Hu, Christoph Feichtenhofer, and Kaiming He. Scaling language-image pre-training via masking. In *Proceedings of the IEEE/CVF conference on computer vision and pattern recognition*, pages 23390–23400, 2023.

Paul Pu Liang, Zihao Deng, Martin Q Ma, James Y Zou, Louis-Philippe Morency, and Ruslan Salakhutdinov. Factorized contrastive learning: Going beyond multi-view redundancy. *Advances in Neural Information Processing Systems*, 36:32971–32998, 2023.

Zhiqiu Lin, Samuel Yu, Zhiyi Kuang, Deepak Pathak, and Deva Ramanan. Multimodality helps unimodality: Cross-modal few-shot learning with multimodal models. In *Proceedings of the IEEE/CVF Conference on Computer Vision and Pattern Recognition*, pages 19325–19337, 2023.

Haotian Liu, Chunyuan Li, Qingyang Wu, and Yong Jae Lee. Visual instruction tuning. *Advances in neural information processing systems*, 36:34892–34916, 2023.

Ming-Yu Liu, Thomas Breuel, and Jan Kautz. Unsupervised image-to-image translation networks. *Advances in neural information processing systems*, 30, 2017.

Ilya Loshchilov and Frank Hutter. Decoupled weight decay regularization. *arXiv preprint arXiv:1711.05101*, 2017.

Jiasen Lu, Dhruv Batra, Devi Parikh, and Stefan Lee. Vilbert: Pretraining task-agnostic visiolinguistic representations for vision-and-language tasks. *Advances in neural information processing systems*, 32, 2019.

Subhransu Maji, Esa Rahtu, Juho Kannala, Matthew Blaschko, and Andrea Vedaldi. Fine-grained visual classification of aircraft. *arXiv preprint arXiv:1306.5151*, 2013.

Mayug Maniparambil, Raiymbek Akshulakov, Yasser Abdelaziz Dahou Djilali, Mohamed El Amine Seddik, Sanath Narayan, Karttikeya Mangalam, and Noel E O'Connor. Do vision and language encoders represent the world similarly? In *Proceedings of the IEEE/CVF Conference on Computer Vision and Pattern Recognition*, pages 14334–14343, 2024.

Sachit Menon and Carl Vondrick. Visual classification via description from large language models. *arXiv preprint arXiv:2210.07183*, 2022.

Jack Merullo, Louis Castricato, Carsten Eickhoff, and Ellie Pavlick. Linearly mapping from image to text space. *arXiv preprint arXiv:2209.15162*, 2022.

David Mizrahi, Roman Bachmann, Oguzhan Kar, Teresa Yeo, Mingfei Gao, Afshin Dehghan, and Amir Zamir. 4m: Massively multimodal masked modeling. *Advances in Neural Information Processing Systems*, 36:58363–58408, 2023.

Luca Moschella, Valentino Maiorca, Marco Fumero, Antonio Norelli, Francesco Locatello, and Emanuele Rodolà. Relative representations enable zero-shot latent space communication. *arXiv preprint arXiv:2209.15430*, 2022.

Maria-Elena Nilsback and Andrew Zisserman. Automated flower classification over a large number of classes. In *2008 Sixth Indian conference on computer vision, graphics & image processing*, pages 722–729. IEEE, 2008.

Antonio Norelli, Marco Fumero, Valentino Maiorca, Luca Moschella, Emanuele Rodola, and Francesco Locatello. Asif: Coupled data turns unimodal models to multimodal without training. *Advances in Neural Information Processing Systems*, 36:15303–15319, 2023.

Omkar M Parkhi, Andrea Vedaldi, Andrew Zisserman, and CV Jawahar. Cats and dogs. In *2012 IEEE conference on computer vision and pattern recognition*, pages 3498–3505. IEEE, 2012.

Sarah Pratt, Ian Covert, Rosanne Liu, and Ali Farhadi. What does a platypus look like? generating customized prompts for zero-shot image classification. In *Proceedings of the IEEE/CVF International Conference on Computer Vision*, pages 15691–15701, 2023.

Alec Radford, Jong Wook Kim, Chris Hallacy, Aditya Ramesh, Gabriel Goh, Sandhini Agarwal, Girish Sastry, Amanda Askell, Pamela Mishkin, Jack Clark, et al. Learning transferable visual models from natural language supervision. In *International conference on machine learning*, pages 8748–8763. PmLR, 2021.

Karsten Roth, Jae Myung Kim, A Koepke, Oriol Vinyals, Cordelia Schmid, and Zeynep Akata. Waffling around for performance: Visual classification with random words and broad concepts. In *Proceedings of the IEEE/CVF International Conference on Computer Vision*, pages 15746–15757, 2023.

Shuvendu Roy, Franklin Ogidi, Ali Etemad, Elham Dolatabadi, and Arash Afkanpour. A shared encoder approach to multimodal representation learning. *arXiv preprint arXiv:2503.01654*, 2025.

Jayoung Ryu, Charlotte Bunne, Luca Pinello, Aviv Regev, and Romain Lopez. Cross-modality matching and prediction of perturbation responses with labeled gromov-wasserstein optimal transport. *arXiv preprint arXiv:2405.00838*, 2024.

Yuyang Shi, Valentin De Bortoli, Andrew Campbell, and Arnaud Doucet. Diffusion schrödinger bridge matching. *Advances in Neural Information Processing Systems*, 36:62183–62223, 2023.

Amanpreet Singh, Ronghang Hu, Vedanuj Goswami, Guillaume Couairon, Wojciech Galuba, Marcus Rohrbach, and Douwe Kiela. Flava: A foundational language and vision alignment model. In *Proceedings of the IEEE/CVF conference on computer vision and pattern recognition*, pages 15638–15650, 2022.

Khurram Soomro, Amir Roshan Zamir, and Mubarak Shah. Ucf101: A dataset of 101 human actions classes from videos in the wild. *arXiv preprint arXiv:1212.0402*, 2012.

Siddharth Srivastava and Gaurav Sharma. Omnivec: Learning robust representations with cross modal sharing. *2024 IEEE/CVF Winter Conference on Applications of Computer Vision (WACV)*, pages 1225–1237, 2023. URL https://api.semanticscholar.org/CorpusID:265128674.

Siddharth Srivastava and Gaurav Sharma. Omnivec2 - a novel transformer based network for large scale multimodal and multitask learning. In *CVPR*, pages 27402–27414, 2024. URL https://doi.org/10.1109/CVPR52733.2024.02588.

Nils Sturma, Chandler Squires, Mathias Drton, and Caroline Uhler. Unpaired multi-domain causal representation learning. *Advances in Neural Information Processing Systems*, 36:34465–34492, 2023.

Subash Timilsina, Sagar Shrestha, and Xiao Fu. Identifiable shared component analysis of unpaired multimodal mixtures. *arXiv preprint arXiv:2409.19422*, 2024.

Chenyu Wang, Sharut Gupta, Xinyi Zhang, Sana Tonekaboni, Stefanie Jegelka, Tommi Jaakkola, and Caroline Uhler. An information criterion for controlled disentanglement of multimodal data. *arXiv preprint arXiv:2410.23996*, 2024.

Johnny Xi, Jana Osea, Zuheng Xu, and Jason S Hartford. Propensity score alignment of unpaired multimodal data. *Advances in Neural Information Processing Systems*, 37:141103–141128, 2024.

Jianxiong Xiao, James Hays, Krista A Ehinger, Aude Oliva, and Antonio Torralba. Sun database: Large-scale scene recognition from abbey to zoo. In *2010 IEEE computer society conference on computer vision and pattern recognition*, pages 3485–3492. IEEE, 2010.

Xiaohua Zhai, Xiao Wang, Basil Mustafa, Andreas Steiner, Daniel Keysers, Alexander Kolesnikov, and Lucas Beyer. Lit: Zero-shot transfer with locked-image text tuning. In *Proceedings of the IEEE/CVF conference on computer vision and pattern recognition*, pages 18123–18133, 2022.

Xiaohua Zhai, Basil Mustafa, Alexander Kolesnikov, and Lucas Beyer. Sigmoid loss for language image pre-training. In *Proceedings of the IEEE/CVF international conference on computer vision*, pages 11975–11986, 2023.

Yiyuan Zhang, Kaixiong Gong, Kaipeng Zhang, Hongsheng Li, Yu Qiao, Wanli Ouyang, and Xiangyu Yue. Meta-transformer: A unified framework for multimodal learning. *arXiv preprint arXiv:2307.10802*, 2023.

Jun-Yan Zhu, Taesung Park, Phillip Isola, and Alexei A Efros. Unpaired image-to-image translation using cycle-consistent adversarial networks. In *Proceedings of the IEEE international conference on computer vision*, pages 2223–2232, 2017.

# Appendix

# A  Further Related Works

**Unpaired Multimodal Learning.**  Unpaired data has long been used for image-to-image [Zhu et al., 2017, Liu et al., 2017, Almahairi et al., 2018, Shi et al., 2023] and text-to-text translation [Lample et al., 2018] . More recently, several works have also proposed learning from unpaired data by inferring coarse- or fine-grained alignments through distribution matching or optimal transport objectives [Xi et al., 2024, Demetci et al., 2022, Ryu et al., 2024]. In contrast, we leverage unpaired data for learning representations without the need for explicit or inferred alignment. [Timilsina et al., 2024, Sturma et al., 2023] theoretically analyze the problem of identifying shared latent components and causal structures in unaligned multimodal mixtures. Most closely related to our work is [Lin et al., 2023], which leverages coarse-grained text data such as class names to improve image classification on CLIP using a shared linear head. Another related line of works [Roth et al., 2023, Pratt et al., 2023, Menon and Vondrick, 2022, Gao et al., 2024] leverage prompting templates and pretrained LLMs to generate descriptive class captions, showing improved image classification performance with CLIP. Nonetheless, these methods operate on CLIP with pre-aligned representation spaces, whereas our approach also learns from unpaired data without assuming prior alignment. Several works have also proposed learning large multitask multimodal models with joint encoders and unified embedding spaces [Srivastava and Sharma, 2024, 2023, Zhang et al., 2023, Girdhar et al., 2022, Geng et al., 2022], often using joint training over separate tasks and/or masked prediction objectives. In a similar vein, [Chada et al., 2023] uses a stage-wise training strategy with both unpaired and paired data, and [Girdhar et al., 2023b] trains a single model across visual modalities. However, most of these methods rely on some amount of paired data for preliminary alignment and then leverage abundant modality-specific unpaired data for further improvement. In contrast, our approach demonstrates that a model can implicitly learn cross-modal correlations from purely unpaired data, without requiring explicit alignment as a prerequisite.

**Multimodal Representation Alignment.**  Our method relies on the notion of shared information and structure between unaligned modalities. Closely related to this are works demonstrating that unimodal representations trained without multimodal data are nevertheless converging. Huh et al. [2024] presents evidence that better-performing language models exhibit increased alignment to self-supervised vision models. Similarly, [Maniparambil et al., 2024] shows a latent space alignment between vision and text encoders across backbones and training paradigms, and uses the CKA metric to connect unaligned encoders zero-shot. Earlier works also note alignment between models trained with different datasets and modalities [Moschella et al., 2022, Norelli et al., 2023]. Several works have also shown that a linear projection or MLP is sufficient to stitch together the latent spaces of pretrained vision and language models [Merullo et al., 2022, Liu et al., 2023, Koh et al., 2023]. Zhai et al. [2022] extends this to training a text encoder to align to a frozen pretrained image model; this method was in turn used to integrate DINOv2, a large self-supervised vision model, with a text encoder [Jose et al., 2024].

# B  Supplementary Experimental Details and Assets Disclosure

## B.1  Assets

We do not introduce new data in the course of this work. Instead, we use publicly available widely used image datasets for the purposes of benchmarking and comparison.

## B.2  Hardware and setup

Each experiment was conducted on 1 NVIDIA Tesla V100 GPUs, each with 32GB of accelerator RAM. The CPUs used were Intel Xeon E5-2698 v4 processors with 20 cores and 384GB of RAM. All experiments were implemented using the PyTorch deep learning framework.

## B.3  Datasets

### B.3.1  Image Classification Benchmarks

We evaluate on the following widely-used classification benchmarks: ImageNet [Deng et al., 2009], StanfordCars [Krause et al., 2013], UCF101 [Soomro et al., 2012], Caltech101 [Fei-Fei et al., 2004],

Table 2: Detailed statistics of the 10 datasets for image classification.

| Dataset | Classes | Train | Val | Test |
|---|---|---|---|---|
| Caltech101 [Fei-Fei et al., 2004] | 100 | 4,128 | 1,649 | 2,465 |
| OxfordPets [Parkhi et al., 2012] | 37 | 2,944 | 736 | 3,669 |
| StanfordCars [Krause et al., 2013] | 196 | 6,509 | 1,635 | 8,041 |
| Oxford Flowers [Nilsback and Zisserman, 2008] | 102 | 4,093 | 1,633 | 2,463 |
| Food101 [Bossard et al., 2014] | 101 | 50,500 | 20,200 | 30,300 |
| FGVCAircraft [Maji et al., 2013] | 100 | 3,334 | 3,333 | 3,333 |
| SUN397 [Xiao et al., 2010] | 397 | 15,880 | 3,970 | 19,850 |
| DTD [Cimpoi et al., 2014] | 47 | 2,820 | 1,128 | 1,692 |
| UCF101 [Soomro et al., 2012] | 101 | 7,639 | 1,898 | 3,783 |
| ImageNet [Deng et al., 2009] | 1,000 | 1.28M | N/A | 50,000 |

Oxford Flowers [Nilsback and Zisserman, 2008], SUN397 [Xiao et al., 2010], DTD [Cimpoi et al., 2014], FGVCAircraft [Maji et al., 2013], OxfordPets [Parkhi et al., 2012], and Food101 [Bossard et al., 2014]. More details about the dataset and splits is provided in Table 2.

### B.3.2 Constructing text templates

To construct conceptually related yet unpaired text data, we generate text templates that capture varying granularities of information about the dataset. Our first approach (*Vanilla*) uses the straightforward template ''a photo of a {}'' with a natural language label for each category, resulting in a basic text description for each class. However, this simple textual corpus lacks fine-grained information necessary to distinguish between visually similar subcategories or to resolve contextually ambiguous terms. To address this, for the second template, we draw from the extensive literature on improving text prompts for zero-shot classification in CLIP [Gao et al., 2024, Menon and Vondrick, 2022, Pratt et al., 2023, Roth et al., 2023]. Specifically, for the second approach (*GPT-3 Descriptions*), we adopt the text prompt generation strategy developed by Pratt et al. [2023], using large language models such as GPT-3 to generate diverse and contextually rich prompts for each image category. We use three generic hand-written sentences across the datasets:

```
Describe what a/the {} looks like:
            Describe a/the {} :
What are the identifying characteristics of a/the {}?
```

The blank portion of each template is populated with the category name, along with the category type for specialized datasets (e.g., "pet" + {} for Oxford Pets or "aircraft" + {} for FGVC Aircraft). The type specification is important for disambiguating categories with multiple interpretations. Some examples of these descriptions are provided in Table 3 for the Oxford Pets dataset.

### B.3.3 ImageNet-ESC Dataset

**Experimental Setup.** We extend our results beyond vision and language to an audiovisual-language dataset: the ImageNet-ESC benchmark [Lin et al., 2023]. This benchmark combines ImageNet (1000 object categories) and ESC-50 (50 environmental sound classes) by matching classes that logically correspond. For example, the dog (barking) class from ESC-50 aligns with various dog breeds from ImageNet, while the clock-alarm sound maps to both analog clock and digital clock. This alignment captures the relationship between visual objects, their sounds, and their textual descriptions. The benchmark consists of two versions: 1) ImageNet-ESC-27: A broader set including loosely matched visual-audio pairs (e.g., drinking-sipping to water bottle); 2) ImageNet-ESC-19: A more precise subset containing only accurate visual-audio matches.

### B.4 Training Protocol

### B.4.1 Image Classification using Image and Unpaired Texts

For text, we use OpenLLaMA-3B as our default encoder and ablate against BERT-Large, RoBERTa-Large, GPT-2 Large, and the pre-aligned CLIP text encoder, keeping the text encoder frozen. For

Table 3: Sample text descriptions per class for Oxford Pets dataset

| Class | Examples |
|---|---|
| Wheaten Terrier | A wheaten terrier is a small, shaggy dog with a soft, silky coat.
A wheaten terrier has a soft, wheat-colored coat that is low-shedding and hypoallergenic.
The wheaten terrier is a medium-sized, hypoallergenic dog breed.
A pet Wheaten Terrier usually has an intelligent expression and a soft, wheat-colored coat. |
| Great Pyrenees | A great pyrenees is a large, white, shaggy-coated dog.
A Great Pyrenees is a large, fluffy dog with a calm, gentle disposition.
The great pyrenees was originally bred to protect livestock from predators.
Great Pyrenees are known for being very large, white dogs with thick fur. |
| Sphynx | A pet Sphynx typically has a small, wrinkled head and a hairless body.
A Sphynx is a hairless cat breed known for its soft, warm skin.
A Sphynx often displays large ears, pronounced cheekbones, and no fur.
Sphynx are unique cats characterized by their lack of coat and wrinkled skin. |
| Birman | A Birman is a long-haired, color-pointed cat with a "mask" of darker fur on its face.
A Birman has silky, pale cream to ivory fur with deep seal- or lilac-colored points.
Birman cats possess striking blue eyes and contrasting white "gloves" on their paws.
They are known for being gentle, affectionate, and smooth-coated companions. |
| Pomeranian | A Pomeranian is a small, fluffy dog with a thick double coat.
Pomeranians are toy-sized, alert dogs with fox-like faces and plumed tails.
A pet Pomeranian often comes in orange, black, white, or mixed coat colors.
They are lively, outgoing, and known for their bold, friendly personalities. |

images, our main backbone is ViT-S/14 DINOv2, with ablations across other DINOv2 variants and the CLIP vision encoder. In the linear-probe setting, all encoder weights stay fixed and we train only a single linear classification head; in full fine-tuning, we jointly update the image backbone and that head, while still freezing the text encoder.

We optimize cross-entropy loss via AdamW [Loshchilov and Hutter, 2017] and perform an extensive grid search over learning rate, weight decay, cosine learning rate scheduling with linear warmup, dropout, and a learnable, modality-specific scaling on the logits. The results are reported for the best-performing model on the validation dataset. We report results for the model achieving highest validation accuracy; the full hyperparameter ranges are in Table 4.

For full fine-tuning, we jointly update the image backbone and classification head with a fixed learning rate of $5 \times 10^{-5}$, batch size 64, and omit learnable modality-specific scaling, since it showed no benefit in this setting.

Table 4: Hyperparameter grid for linear probing.

| Hyperparameter | Values |
|---|---|
| Optimizer | adamw |
| Learning rate | {0.001, 1e-4} |
| Weight decay | {0.0, 0.01, 0.001} |
| LR scheduler | cosine |
| Batch size | {8, 32} |
| Max iterations | 12,800 |
| Warmup iterations | 50 |
| Warmup type | linear |
| Warmup min LR | 1e-5 |
| Dropout | {0.0} |
| Modality-specific learnable scaling | {False, True} |
| Early-stop patience | 10 |

### B.4.2 Evaluation on ImageNet-ESC

Similar to our vision-language experiments, we perform few-shot evaluation using the 5-fold splits defined in the benchmark. Each fold contains 8 samples per class, with one fold used for training and validation and the remaining four for testing. We repeat the process over 5 random splits and report the average performance. For audio encoding, we use AudioCLIP with an ES-ResNeXT backbone [Guzhov et al., 2021]. AudioCLIP is pretrained on AudioSet and generates audio embeddings in the same representation space as CLIP. Following the instructions in [Guzhov et al., 2021, Lin et al., 2023], we use `train()` mode in Pytorch to extract the features since `eval()` mode yields suboptimal embeddings. We evaluate our models on two tasks—audio classification and image classification—comparing the unimodal baseline against two multimodal variants in which the primary modality is each time augmented by one of the other modalities.

### B.4.3 Transfer Learning from Language to Vision

To adapt a language model to image classification, we embed image patches using a linear projection and add positional encodings to capture spatial structure. We then use transformer layers initialized from pretrained BERT, and finally, a 2-layer MLP classification head. Specifically, we split each image of size $224 \times 224$ into patches of size $16 \times 16$ with 196 patch tokens. Each patch is then projected into the model's embedding space of dimension $d$(*e.g.* d=768 for GPT-2, $d = 1024$ for BERT) via a learned linear layer. We then prepend a learnable "[CLS]" token, add learned positional embeddings of shape $(N + 1) \times d$, and apply dropout with probability $p = 0.1$. This $(N + 1) \times d$ sequence is passed into the pretrained transformer stack (either GPT-2 or BERT), using a full bidirectional attention mask over all patch tokens and the CLS token. We extract the final hidden state corresponding to the CLS token and feed it through a two-layer MLP classification head.

During training, we evaluate two scenarios: 1) one where the pretrained backbone is frozen and only the patch embedding and linear head are trained, and 2) another where the backbone is initially frozen to align the trainable layers (patch embedding and head) with the pretrained language backbone, and then unfrozen after 2000 steps for end-to-end training. This approach allows us to test whether the semantic richness captured by language models provides a strong initialization, leading to better convergence and performance compared to training ViT from scratch.

## C Proofs of Theoretical Results

In this section, we present complete derivations and proofs of the main theoretical claims. Appendix C.1 gathers all definitions and background required for our arguments. Appendix C.2 formalizes the linear data-generating model, derives closed-form maximum-likelihood estimators for each modality and their joint estimator, and computes the corresponding block-wise Fisher information. Finally, Appendix C.3 provides the detailed proofs of our variance-reduction claims, showing rigorously how unpaired multimodal estimation strictly lowers estimator variance.

### C.1 Background and Definitions

In this section we revisit the mathematical definitions used in our theoretical analysis, including matrix-orderings, characterization of symmetric matrices and Fisher information.

**Definition 1** (Positive Semidefinite Matrix). A real symmetric matrix $A \in \mathbb{R}^{d \times d}$ is *positive semidefinite* if for all vectors $v \in \mathbb{R}^d$, $v^\top A v \geq 0$. Equivalently, all eigenvalues of $A$ are nonnegative. We denote the set of all $d \times d$ symmetric, positive-semidefinite matrices as $S^d_{\succeq 0}$.

**Definition 2** (Positive Definite Matrix). A real symmetric matrix $A \in \mathbb{R}^{d \times d}$ is *positive definite* if for every nonzero $v \in \mathbb{R}^d$, $v^\top A v > 0$. Equivalently, all eigenvalues of $A$ are strictly positive. We denote the set of all $d \times d$ symmetric, positive definite matrices as $S^d_{\succ 0}$.

**Definition 3** (Loewner Order). For two real symmetric matrices $A, B \in \mathbb{R}^{d \times d}$, we write $A \preceq B \iff B - A$ is positive semidefinite and $A \prec B \iff B - A$ is positive definite. This defines a partial order on the cone of symmetric matrices.

**Definition 4** (Fisher Information Matrix). Given a parametric family of densities $p(x; \theta)$ on data $x$, the *Fisher information matrix* at parameter $\theta$ is

$$I(\theta) = \mathbb{E}_{x \sim p(\cdot; \theta)}\left[\nabla_\theta \log p(x; \theta) \nabla_\theta \log p(x; \theta)^\top\right].$$

Equivalently, for regular models, $I(\theta) = -\mathbb{E}\left[\nabla_\theta^2 \log p(x; \theta)\right]$.

## C.2  Maximum Likelihood Estimators and Fisher Contributions

In this section we revisit our linear data–generating model, introduce notations for the $X$–only, $Y$–only and joint likelihoods, derive the closed-form MLEs $\widehat{\theta}_X$, $\widehat{\theta}_Y$ and $\widehat{\theta}_{X,Y}$, and formalize their information contributions towards estimating the ground truth parameters $\theta \equiv [\theta_c, \theta_x, \theta_y]^\top$.

**Data Generating Process.** Recall our linear data-generating process: Assume that all factors of variation in reality live in a single $d$-dimensional space $\mathcal{Z}^* \equiv \theta \in \mathbb{R}^d$ modeled using a linear data-generating pipeline. This parameter can further be decomposed as $\theta \equiv [\theta_c, \theta_x, \theta_y]^\top$ where $\theta_c \in \mathbb{R}^{d_c}, \theta_x \in \mathbb{R}^{d_x}, \theta_y \in \mathbb{R}^{d_y}$ and $d_c + d_x + d_y = d$. Here, $\theta_c$ captures the *common* (shared) parameters that affect both modalities, $\theta_x$ denotes the parameters that only affect modality $X$, and $\theta_y$ denotes the parameters that only affect modality $Y$. We observe two independent datasets, one from each modality $\{X_i\}_{i=1}^{N_x} \in \mathbb{R}^m$ and $\{Y_j\}_{j=1}^{N_y} \in \mathbb{R}^n$, each reflecting partial measurements of the ground truth latent space $\mathcal{Z}^*$:

$$X_i = A_{c,i}\, \theta_c + A_{x,i}\, \theta_x + \epsilon_{X,i}, \quad \epsilon_{X,i} \sim \mathcal{N}\left(0, \sigma_x^2 I_{m_i}\right) \tag{1}$$

$$Y_j = B_{c,j}\, \theta_c + B_{y,j}\, \theta_y + \epsilon_{Y,j}, \quad \epsilon_{Y,j} \sim \mathcal{N}\left(0, \sigma_y^2 I_{n_j}\right). \tag{2}$$

Here, $A_{c,i}, A_{x,i}, B_{c,j}, B_{y,j}$ are known design blocks capturing how each sample probes the latent factors and $\varepsilon_{X,i}, \varepsilon_{Y,j}$ represent the independent measurement noise.

In our linear setting, estimating the true latent state $\theta$—and hence the underlying reality $\mathcal{Z}^*$—is governed by the Fisher information matrix $I(\theta) = -\mathbb{E}\left[\nabla_\theta^2 \ell(\theta)\right]$, which measures how sharply the likelihood "curves" around the true $\theta$. High curvature along a particular axis means the data tightly constrain that component, driving down estimator variance there.

**Unimodal Estimators.** We first estimate $\theta$ using only the $X$–dataset. Stacking $\{X_i\}_{i=1}^{N_x}$ yields a design matrix $\mathcal{A}$ with block rows $[A_{c,i},\ A_{x,i},\ 0]$. The least-squares solution

$$\widehat{\theta}_X = \arg\min_\theta \sum_{i=1}^{N_x} \left\|X_i - A_{c,i}\, \theta_c - A_{x,i}\, \theta_x\right\|^2$$

omits $\theta_y$ entirely. Consequently, the Fisher information on $\theta_y$ vanishes, making it unidentifiable.

Analogously, stacking $\{Y_j\}_{j=1}^{N_y}$ defines $\mathcal{B}$ with block rows $[B_{c,j},\ 0,\ B_{y,j}]$ and yields

$$\widehat{\theta}_Y = \arg\min_\theta \sum_{j=1}^{N_y} \left\|Y_j - B_{c,j}\, \theta_c - B_{y,j}\, \theta_y\right\|^2.$$

This estimator doesn't depend on $\theta_x$, providing zero coverage for that component. Thus, each unimodal estimator entirely fails to recover the parameters exclusive to the omitted modality.

**Multimodal Estimators.** Despite the lack of one-to-one pairing, both $\{X_i\}$ and $\{Y_j\}$ share the common parameters $\theta_c$. Since the two distributions are independent, the joint likelihood factorizes as

$$\prod_{i=1}^{N_x} p(X_i \mid \theta_c, \theta_x) \times \prod_{j=1}^{N_y} p(Y_j \mid \theta_c, \theta_y).$$

Maximizing this yields the combined estimator

$$\widehat{\theta}_{X,Y} = \arg\min_{\theta_c, \theta_x, \theta_y} \left\{ \sum_{i=1}^{N_x} \|X_i - A_{c,i}\, \theta_c - A_{x,i}\, \theta_x\|^2 + \sum_{j=1}^{N_y} \|Y_j - B_{c,j}\, \theta_c - B_{y,j}\, \theta_y\|^2 \right\}.$$

Intuitively, there is no requirement to match up individual $(X_i, Y_j)$ pairs. Instead, the estimate for $\theta_c$ is improved by both modalities while remaining unpaired.

**Fisher Information.** In our linear model, each dataset contributes block-structured Fisher information. For the $X$–dataset:

$$I_X = \sum_{i=1}^{N_x} \begin{pmatrix} A_{c,i}^\top A_{c,i} & A_{c,i}^\top A_{x,i} & 0 \\ A_{x,i}^\top A_{c,i} & A_{x,i}^\top A_{x,i} & 0 \\ 0 & 0 & 0 \end{pmatrix},$$

and for the $Y$–dataset:

$$I_Y = \sum_{j=1}^{N_y} \begin{pmatrix} B_{c,j}^\top B_{c,j} & 0 & B_{c,j}^\top B_{y,j} \\ 0 & 0 & 0 \\ B_{y,j}^\top B_{c,j} & 0 & B_{y,j}^\top B_{y,j} \end{pmatrix}.$$

Because $X$ and $Y$ samples are independent, their curvature contributions add pointwise, resulting in the joint Fisher information being simply the sum of the unimodal blocks.

$$I_{X,Y} = I_X + I_Y = \begin{pmatrix} \sum_i A_{c,i}^\top A_{c,i} + \sum_j B_{c,j}^\top B_{c,j} & * & * \\ * & \sum_i A_{x,i}^\top A_{x,i} & 0 \\ * & 0 & \sum_j B_{y,j}^\top B_{y,j} \end{pmatrix},$$

where "$*$" denotes the cross-modal blocks. In particular, we have the shared-parameter block as

$$(I_{X,Y})_{\theta_c, \theta_c} = \sum_{i=1}^{N_x} A_{c,i}^\top A_{c,i} + \sum_{j=1}^{N_y} B_{c,j}^\top B_{c,j},$$

## C.3 Theorems and Proofs

The aim of this section is to detail the proofs of the theoretical results presented in the main manuscript The key theoretical tools driving our analysis are already prepared in Appendix C.1 and Appendix C.2. Core to our theoretical analysis are a few lemmas around the Loewner-order monotonicity result for inverses that we prove below.

**Lemma 1** (Loewner Order reversal for inverses). *Let $M, N \in \mathbb{S}_{\succ 0}^d$ with $M \prec N$ (or $M \preceq N$). Then $N^{-1} \prec M^{-1}$ (or $N^{-1} \preceq M^{-1}$).*

*Proof.* Since $N \succ 0$, $N^{-1/2}$ exists and is nonsingular. Define $C := N^{-1/2} M N^{-1/2} \prec I$. Because a congruence with an invertible matrix preserves positive-definiteness, $C \succ 0$; hence $C^{-1}$ is well defined and $C^{-1} \succ I$ (the scalar map $x \mapsto x^{-1}$ is strictly decreasing on $(0, \infty)$). Undoing the congruence gives

$$M^{-1} = N^{-1/2} C^{-1} N^{-1/2} \succ N^{-1/2} I N^{-1/2} = N^{-1}. \qquad \square$$

**Lemma 2** (Inverse–monotonicity of the Moore–Penrose pseudoinverse). *Let $M, N \in \mathbb{S}_{\succeq 0}^d$ satisfy $M \prec N$ and $\ker M = \ker N =: K$. Then their pseudoinverses obey $N^\dagger \prec M^\dagger$.*

*Proof.* Set $S := K^\perp$ and let $P := P_S$ be the orthogonal projector onto $S$. Because $M$ and $N$ vanish on $K$, we have the decompositions $M = PMP$ and $N = PNP$. Restricted to $S$ both matrices are positive–definite:

$$\tilde{M} := PMP, \qquad \tilde{N} := PNP \in \mathbb{S}_{\succ 0}^{\dim S}, \quad \tilde{M} \prec \tilde{N}.$$

Apply Lemma 1 to $\tilde{M}, \tilde{N}$ to obtain $\tilde{N}^{-1} \prec \tilde{M}^{-1}$ on $S$. The Moore–Penrose pseudoinverse equals the ordinary inverse on $S$ and is zero on $K$:

$$M^\dagger = P \tilde{M}^{-1} P, \qquad N^\dagger = P \tilde{N}^{-1} P.$$

Therefore $N^\dagger = P \tilde{N}^{-1} P \prec P \tilde{M}^{-1} P = M^\dagger$. $\qquad \square$

**Lemma 3** (Directional Loewner Order reversal). *Let $M, N \in \mathbb{S}_{\succ 0}^d$ with $M \preceq N$. If a non-zero vector $v$ satisfies $v^\top M v < v^\top N v$, then*

1. *For the vector $v$, it holds that $v^\top M^{-1}v \geq v^\top N^{-1}v$, with strict inequality $v^\top M^{-1}v > v^\top N^{-1}v$ if and only if $(N - M)M^{-1}v \neq 0$.*

2. *There exists a non-zero vector $u \in \mathbb{R}^d$ such that $u^\top M^{-1}u > u^\top N^{-1}u$.*

*Proof.* Denote the Loewner gap $\Delta := N - M \succeq 0$. Then, the assumption $v^\top Nv > v^\top Mv$ is equivalent to $v^\top \Delta v > 0$. Introduce the congruence–invariant normalisation $C := M^{-1/2}\Delta M^{-1/2} \succeq 0$. Now, using $\Delta = M^{1/2}CM^{1/2}$ and properties of inverse,

$$N = M^{1/2}(I + C)M^{1/2}, \qquad N^{-1} = M^{-1/2}(I + C)^{-1}M^{-1/2},$$

since $I + C \succ 0$ (because $C \succeq 0$ and $I \succ 0$). Thus,

$$M^{-1} - N^{-1} = M^{-1/2}\Big[I - (I + C)^{-1}\Big]M^{-1/2}$$
$$= M^{-1/2}C(I + C)^{-1}M^{-1/2},$$

because $(I - (I + C)^{-1})(I + C) = C$. Finally, evaluating in the direction $v$, we have

$$v^\top(M^{-1} - N^{-1})v = v^\top M^{-1/2}(I + C)^{-1}CM^{-1/2}v$$
$$= u^\top(I + C)^{-1}Cu \quad \text{(where } u = M^{-1/2}v)$$

Now, since $(I + C)^{-1} \in \mathbb{S}_{\succ 0}$ and $C \in \mathbb{S}_{\succeq 0}$ commute, the matrix $(I + C)^{-1}C$ is positive semidefinite and it has exactly the same kernel as $C$. Thus, if $C = Q\operatorname{diag}(\lambda_i)Q^\top$ ($\lambda_i \geq 0$), we have

$$u^\top C(I + C)^{-1}u = \sum_i \frac{\lambda_i}{1 + \lambda_i}(Q^\top u)_i^2 \geq 0.$$

This expression is strictly positive exactly when $u$ has a component in any eigen-subspace with $\lambda_i > 0$ i.e when $u \notin \ker(C)$. Since $M^{-1/2} \in \mathbb{S}_{\succ 0}$, $Cu = 0 \implies M^{-1/2}\Delta M^{-1/2}u = 0 \implies \Delta M^{-1}v = 0$. Thus, this expression is strictly positive if $\Delta M^{-1}v \neq 0$.

Now, from the premise $v^\top \Delta v > 0$, it follows that $\Delta \neq 0$. Since $M \succ 0$, $M^{-1/2}$ is invertible, $C$ is also not the zero matrix. Since $C \succeq 0$, this means that $C$ must have at least one strictly positive eigenvalue. Let $\lambda > 0$ be such an eigenvalue, and let $z \neq 0$ be a corresponding eigenvector. Define, $x := M^{1/2}z \neq 0$. Thus, we have $x^\top(M^{-1} - N^{-1})x = z^\top C(I + C)^{-1}z = \frac{\lambda}{1+\lambda}\|z\|^2 > 0$, showing the existence of a non-zero vector $x$ such that $x^\top M^{-1}x > x^\top N^{-1}x$.

$\square$

**Theorem 1.** *Let $\hat{\theta}_X, \hat{\theta}_Y$ be the least-squares estimators for $\theta$ using only $\{X_i\}$ and only $\{Y_j\}$ and let $\hat{\theta}_{X,Y}$ be the joint estimator using both unpaired datasets. Then, under the assumption that at least one $B_{c,j}$ where $j \in \{1, 2, ...N_y\}$ has full rank, the common-factor covariance satisfies the strict Loewner ordering i.e. $\operatorname{Var}(\hat{\theta}_{X,Y})_{\theta_c,\theta_c} \prec \operatorname{Var}(\hat{\theta}_X)_{\theta_c,\theta_c}$, or equivalently, the Fisher information on $\theta_c$ strictly increases when combining both modalities, despite not having sample-wise pairing: $(I_X + I_Y)_{\theta_c,\theta_c} \succ (I_X)_{\theta_c,\theta_c}$.*

*Proof.* For any statistic $S(\theta) = \nabla_\theta \log p(x; \theta)$ and vector $v$,

$$v^\top I(\theta)\,v = v^\top \mathbb{E}[S(\theta)S(\theta)^\top]\,v = \mathbb{E}[(v^\top S(\theta))^2] \geq 0.$$

Thus, a Fisher Information Matrix is a positive semidefinite matrix.

In our linear–Gaussian model, the $X$–dataset contributes $(I_X)_{\theta_c,\theta_c} = \sum_{i=1}^{N_x} A_{c,i}^\top A_{c,i}$ and the $Y$–dataset gives $(I_Y)_{\theta_c,\theta_c} = \sum_{j=1}^{N_y} B_{c,j}^\top B_{c,j}$. Since at least one $B_{c,j}$ has full column rank, $(I_Y)_{\theta_c,\theta_c}$ is positive-definite on the $\theta_c$ subspace. Now, if at least one $B_{c,j} \in \mathbb{R}^{m \times d_c}$ has full column rank $d_c$, then for any $v \in \mathbb{R}^{d_c} \setminus \{0\}$,

$$v^\top B_{c,j}^\top B_{c,j}\,v = \|B_{c,j}v\|^2 > 0.$$

Hence, each summand in $(I_Y)_{\theta_c,\theta_c}$ is positive semidefinite and at least one is positive definite, so their sum $\sum_j B_{c,j}^\top B_{c,j}$ is positive definite on the $\theta_c$ subspace. Thus,

$$(I_X)_{\theta_c,\theta_c} \prec (I_X)_{\theta_c,\theta_c} + (I_Y)_{\theta_c,\theta_c} = (I_X + I_Y)_{\theta_c,\theta_c}$$

Now, for regular exponential families (including Gaussian linear models), the covariance matrix of the maximum likelihood estimator $\widehat{\theta}$ near the true $\theta_0$ is (asymptotically) the inverse of the Fisher information matrix i.e. $\mathrm{Var}(\widehat{\theta}) \approx I(\theta_0)^{-1}$. Precisely, as the sample size $n \to \infty$, we have:

$$\sqrt{n}(\hat{\theta} - \theta_0) \xrightarrow{d} \mathcal{N}(0, I(\theta_0)^{-1}),$$

where $\theta_0$ is the true parameter value, $I(\theta_0)$ is the Fisher Information Matrix evaluated at $\theta_0$ and $\mathcal{N}(0, I(\theta_0)^{-1})$ denotes a multivariate normal distribution with mean 0 and covariance matrix $I(\theta_0)^{-1}$. Thus, we compare variances via the Moore–Penrose pseudoinverse of the information matrices.

Let $M_X = (I_X)_{\theta_c,\theta_c}$, $M_Y = (I_Y)_{\theta_c,\theta_c}$ and $M_{X,Y} = (I_X + I_Y)_{\theta_c,\theta_c}$. Since $M_Y \succ 0$, $M_{X,Y} = M_X + M_Y$ is also positive definite (as $M_{X,Y} \succeq M_Y \succ 0$). Thus, $\mathrm{Var}(\hat{\theta}_{X,Y}) = M_{X,Y}^{-1}$. We have established $M_X \prec M_{X,Y}$. Assuming $M_X$ is positive definite (to define the matrix $\mathrm{Var}(\hat{\theta}_{X,Y})_{\theta_c,\theta_c}$), we apply Lemma 1 to get $M_{X,Y}^{-1} \prec M_X^{-1}$. Thus,

$$\mathrm{Var}(\hat{\theta}_{X,Y})_{\theta_c,\theta_c} = M_{X,Y}^{-1} \prec M_X^{-1} = \mathrm{Var}(\hat{\theta}_X)_{\theta_c,\theta_c},$$

This proves the statement under the condition that $M_X$ is positive definite. Note here that, on spaces unidentifiable by $X$-alone i.e. $v \in \ker(M_X)$, we have $\mathrm{Var}(\hat{\theta}_X)_{\theta_c,\theta_c} = \infty$. Since $M_{X,Y}$ is positive definite, it has finite variance along such $v$ i.e. $\mathrm{Var}(\hat{\theta}_{X,Y})_{\theta_c,\theta_c} < \infty$, thus strictly reducing the variance of the estimator. Thus, adding the unpaired $Y$-modality strictly reduces the variance (or, dually, increases the Fisher information) on the common factors $\theta_c$.

$\square$

**Theorem 2.** *Let all notation be as in Theorem 1, and define $M_X := (I_X)_{\theta_c,\theta_c}$, $M_Y := (I_Y)_{\theta_c,\theta_c}$, and $M_{XY} := M_X + M_Y$. Let $v \in \mathbb{R}^{d_c} \setminus \{0\}$. If there exists at least one index $j \in \{1, 2, ... N_y\}$ such that $B_{c,j}v \neq 0$, then the following hold:*

1. *The Fisher information strictly increases in direction $v$ i.e. $v^\top M_{XY}\, v > v^\top M_X v$.*

2. *The variance of the estimator in direction $v$ is strictly reduced i.e $v^\top \mathrm{Var}(\hat{\theta}_{X,Y})_{\theta_c,\theta_c}\, v < v^\top \mathrm{Var}(\hat{\theta}_X)_{\theta_c,\theta_c}\, v$, if $v \notin \mathrm{range}(M_X)$. For $v \in \mathrm{range}(M_X)$, this strict inequality holds for $v$ under an additional invertibility condition and is always guaranteed for some $u \in \mathrm{range}(M_X)$ i.e. $\exists u$ s.t. $u^\top \mathrm{Var}(\hat{\theta}_{X,Y})_{\theta_c,\theta_c}\, u < u^\top \mathrm{Var}(\hat{\theta}_X)_{\theta_c,\theta_c}\, u$.*

*Proof.* Define $M_X := (I_X)_{\theta_c,\theta_c}$, $M_Y := (I_Y)_{\theta_c,\theta_c}$, $and M_{XY} := M_X + M_Y$. By assumption, $\exists j$ such that $B_{c,j}v \neq 0$. Thus:

$$v^\top M_Y v = \sum_{j=1}^{N_y} \|B_{c,j}v\|^2 \geq \|B_{c,j}v\|^2 > 0.$$

Hence $M_Y$ is positive-definite in direction $v$, implying $M_{X,Y} \succ M_X$ in this direction:

$$v^\top M_{XY}v = v^\top M_X v + v^\top M_Y v > v^\top M_X v,$$

thus proving the first part of the theorem.

**Case 1:** $v \notin \mathrm{Range}(M_X)$. If $v \notin \mathrm{Range}(M_X)$, then $v$ has a non-zero component in $\ker(M_X)$. Let $v = v_S + v_K$, where $v_S \in \mathrm{Range}(M_X)$ and $v_K \in \ker(M_X)$ with $v_K \neq 0$. The linear combination of parameters $v^\top \theta_c = v_S^\top \theta_c + v_K^\top \theta_c$. Since $v_K \in \ker(M_X)$, the component $v_K^\top \theta_c$ is not identifiable by the $X$-only model. Consequently, the asymptotic variance of an unbiased estimator for $v^\top \theta_c$ using only the $X$-dataset is infinite. We denote this as $v^\top \mathrm{Var}(\hat{\theta}_X)_{\theta_c,\theta_c} v = \infty$.

The strict inequality $v^\top M_{XY}v > 0$, ensures that $v \notin \ker(M_{XY})$, and thus $v \in \mathrm{Range}(M_{XY})$. Since $v \in \mathrm{Range}(M_{XY})$ and $v \neq 0$, $M_{XY}^\dagger v$ is well-defined. Furthermore, because $M_{XY}$ is positive

semidefinite, $M_{XY}^\dagger$ is also positive semidefinite and shares the same kernel as $M_{XY}$ (since $M_{XY}$ is symmetric). As $v \neq 0$ and $v \notin \ker(M_{XY})$, thus $v \notin \ker(M_{XY}^\dagger)$, which ensures $v^\top M_{XY}^\dagger v$ is a finite positive value. Thus,

$$v^\top \mathrm{Var}(\hat{\theta}_{X,Y})_{\theta_c,\theta_c} v < \infty.$$

Comparing this to the variance from the $X$-only model in this case:

$$v^\top \mathrm{Var}(\hat{\theta}_{X,Y})_{\theta_c,\theta_c} v < \infty = v^\top \mathrm{Var}(\hat{\theta}_X)_{\theta_c,\theta_c} v,$$

and the strict inequality holds.

**Case 2:** $v \in \mathrm{Range}(M_X)$. Let $S := \mathrm{Range}(M_X)$ and let $P_S$ be the orthogonal projector onto $S$. Because $M_X = M_X P_S$ and $M_{XY} = M_X + M_Y$, the restrictions

$$\tilde{M}_X := P_S M_X P_S, \qquad \tilde{M}_{XY} := P_S M_{XY} P_S = \tilde{M}_X + P_S M_Y P_S$$

are *positive-definite* on $S$; To see this, take any non-zero $w \in S$. Since $w \in \mathrm{range}(M_X)$, $P_S w = w$; hence

$$w^\top \tilde{M}_X w = w^\top M_X w > 0 \quad (P_S \text{ is identity when restricted to } S)$$

Thus $\tilde{M}_X \succ 0$ on $S$. Because $P_S M_Y P_S \succeq 0$, adding it preserves positive-definiteness, so

$$\tilde{M}_{XY} = \tilde{M}_X + P_S M_Y P_S \succeq \tilde{M}_X \succ 0 \quad \text{on } S.$$

Applying Lemma 3(1) to $\tilde{M}_X$ and $\tilde{M}_{XY}$ on $S$ gives us $v^\top \tilde{M}_{XY}^{-1} v \leq v^\top \tilde{M}_X^{-1} v$. Strict inequality $v^\top \tilde{M}_{XY}^{-1} v < v^\top \tilde{M}_X^{-1} v$ holds if and only if the condition $C_v := ((\tilde{M}_{XY} - \tilde{M}_X)\tilde{M}_X^{-1} v \neq \mathbf{0})$ is met. Therefore, if condition $C_v$ holds, the directional variance along this constrained space $S$ is strictly reduced:

$$v^\top \mathrm{Var}(\hat{\theta}_{X,Y})_{\theta_c,\theta_c} v = v^\top \tilde{M}_{XY}^{-1} v < v^\top \tilde{M}_X^{-1} v = v^\top \mathrm{Var}(\hat{\theta}_X)_{\theta_c,\theta_c} v^1.$$

Further, from Lemma 3(2), there exists some non-zero vector $u \in S$ such that $u^\top \tilde{M}_{XY}^{-1} u < u^\top \tilde{M}_X^{-1} u$. Thus we have,

$$u^\top \mathrm{Var}(\hat{\theta}_{X,Y})_{\theta_c,\theta_c} u < u^\top \mathrm{Var}(\hat{\theta}_X)_{\theta_c,\theta_c} u.$$

Thus, completing the proof.

$\square$

**Corollary 1.** *Assume a direction* $v \in \mathbb{R}^{d_c} \setminus \{0\}$ *with* $a = v^\top (I_X)_{\theta_c,\theta_c} v > 0$ *and* $b = v^\top (I_Y)_{\theta_c,\theta_c} v > 0$ *where* $v$ *is the common eigenvector of* $(I_X)_{\theta_c,\theta_c}$ *and* $(I_Y)_{\theta_c,\theta_c}$. *Then the variance in direction* $v$ *contracts by the factor*

$$\frac{v^\top \mathrm{Var}(\hat{\theta}_{X,Y}) v}{v^\top \mathrm{Var}(\hat{\theta}_X) v} = \frac{1/(a+b)}{1/a} = \frac{a}{a+b} < 1,$$

*So the joint estimator achieves strictly lower error along* $v$.

*Proof.* Let $M_X = (I_X)_{\theta_c,\theta_c}$ and $M_Y = (I_Y)_{\theta_c,\theta_c}$. By assumption, $v$ is a common eigenvector of $M_X$ and $M_Y$. Thus, $M_X v = \lambda_X v$ and $M_Y v = \lambda_Y v$ for some eigenvalues $\lambda_X$ and $\lambda_Y$. From the assumptions, we have $\lambda_X = a/\|v\|^2 > 0$ and $\lambda_Y = b/\|v\|^2 > 0$. Since $M_X$ is symmetric and $M_X v = \lambda_X v$ with $\lambda_X > 0$, the pseudoinverse acts as $M_X^\dagger v = \lambda_X^{-1} v$. Therefore, the variance in direction $v$ for the $X$-only estimator is

$$v^\top \mathrm{Var}(\hat{\theta}_X)_{\theta_c,\theta_c} v = v^\top M_X^\dagger v = v^\top (\lambda_X^{-1} v) = \lambda_X^{-1} \|v\|^2 = a^{-1} \|v\|^4.$$

Since $v$ is a common eigenvector, it is also an eigenvector of $M_{XY} = M_X + M_Y$:

$$(M_X + M_Y)v = M_X v + M_Y v = \lambda_X v + \lambda_Y v = (\lambda_X + \lambda_Y)v.$$

---

[1]We note that true asymptotic variance defined as $v^\top \mathrm{Var}(\hat{\theta}_{X,Y})_{\theta_c,\theta_c} v = v^\top M_{XY}^\dagger v$, $v^\top M_{XY}^\dagger v = v^\top \tilde{M}_{XY}^{-1} v$ if $S$ is an invariant subspace of $M_{XY}$ and $M_{XY}$ is block-diagonal with respect to $S$ and $S^\perp$ (i.e., $P_S M_{XY} P_{S^\perp} = \mathbf{0}$, which implies $P_S M_Y P_{S^\perp} = \mathbf{0}$).

The corresponding eigenvalue is $\lambda_{XY} = \lambda_X + \lambda_Y$. Since $\lambda_X > 0$ and $\lambda_Y > 0$, $\lambda_{XY} > 0$. Thus, $(M_X + M_Y)^\dagger v = (\lambda_X + \lambda_Y)^{-1} v$. The variance in direction $v$ for the joint estimator is

$$v^\top \mathrm{Var}(\hat{\theta}_{X,Y})_{\theta_c,\theta_c} \, v = v^\top (M_X + M_Y)^\dagger v = (\lambda_X + \lambda_Y)^{-1} \|v\|^2 = (a+b)^{-1} \|v\|^4.$$

Now, we form the ratio of these variances:

$$\frac{v^\top \mathrm{Var}(\hat{\theta}_{X,Y})_{\theta_c,\theta_c} \, v}{v^\top \mathrm{Var}(\hat{\theta}_X)_{\theta_c,\theta_c} \, v} = \frac{\lambda_X}{\lambda_X + \lambda_Y} = \frac{a}{a+b} < 1.$$

$\square$

**Corollary 2.** *Assume a direction $v \in \mathbb{R}^{d_c} \setminus \{0\}$ with $v^\top (I_X)_{\theta_c,\theta_c} \, v = 0$ and $v^\top (I_Y)_{\theta_c,\theta_c} \, v > 0$. Then $v^\top \mathrm{Var}(\hat{\theta}_X) \, v = \infty$ and $v^\top \mathrm{Var}(\hat{\theta}_{X,Y}) \, v < \infty$ i.e. a direction unidentifiable from $X$ alone becomes well-posed with even unpaired data from $Y$.*

*Proof.* This corollary follows directly from Case 1 of Theorem 2. The condition $v^\top (I_X)_{\theta_c,\theta_c} \, v = 0$ for $v \neq 0$ implies $v \in \ker((I_X)_{\theta_c,\theta_c})$, and thus $v \notin \mathrm{range}((I_X)_{\theta_c,\theta_c})$. Given the additional condition $v^\top (I_Y)_{\theta_c,\theta_c} \, v > 0$, the conclusions of Case 1 of the theorem apply directly. $\square$

**Corollary 3** (Variance Reduction for Eigenvectors of $M_X$). *Let $v \in \mathbb{R}^{d_c} \setminus \{0\}$ be an eigenvector of $M_X = (I_X)_{\theta_c,\theta_c}$ with a corresponding eigenvalue $\lambda_X > 0$. If the $Y$-dataset provides information in this direction $v$ (i.e., $v^\top M_Y v > 0$, where $M_Y = (I_Y)_{\theta_c,\theta_c}$), then the variance in direction $v$ is strictly reduced by incorporating the $Y$-dataset:*

$$v^\top \mathrm{Var}(\hat{\theta}_{X,Y})_{\theta_c,\theta_c} \, v < v^\top \mathrm{Var}(\hat{\theta}_X)_{\theta_c,\theta_c} \, v.$$

*Specifically, $v^\top \mathrm{Var}(\hat{\theta}_X)_{\theta_c,\theta_c} v = \lambda_X^{-1} \|v\|^2$.*

*Proof.* Let $M_X = (I_X)_{\theta_c,\theta_c}$ and $M_Y = (I_Y)_{\theta_c,\theta_c}$. Since $v$ is an eigenvector of $M_X$ with a positive eigenvalue $\lambda_X > 0$, it follows that $v \in \mathrm{Range}(M_X)$. Let $S = \mathrm{Range}(M_X)$. The variance using only the $X$-dataset in direction $v$ is given by

$$v^\top \mathrm{Var}(\hat{\theta}_X)_{\theta_c,\theta_c} v = v^\top M_X^\dagger v.$$

Because $v$ is an eigenvector of $M_X$ with $\lambda_X > 0$, $M_X^\dagger v = \lambda_X^{-1} v$. Thus,

$$v^\top \mathrm{Var}(\hat{\theta}_X)_{\theta_c,\theta_c} v = v^\top (\lambda_X^{-1} v) = \lambda_X^{-1} \|v\|^2.$$

This scenario falls under Case 2 of Theorem 2, specifically its conclusion regarding $v \in S$. According to that theorem, strict variance reduction $v^\top \mathrm{Var}(\hat{\theta}_{X,Y})_{\theta_c,\theta_c} v < v^\top \mathrm{Var}(\hat{\theta}_X)_{\theta_c,\theta_c} v$ occurs if the condition $C_v = ((P_S M_Y P_S)(M_X|_S)^{-1} v \neq \mathbf{0})$ holds. Here, $P_S$ is the orthogonal projector onto $S$, and $M_X|_S$ is the restriction of $M_X$ to $S$, so $(M_X|_S)^{-1} v = \lambda_X^{-1} v$.

The condition $C_v$ thus becomes $(P_S M_Y P_S)(\lambda_X^{-1} v) \neq \mathbf{0}$. Since $\lambda_X > 0$, this is equivalent to $P_S M_Y P_S v \neq \mathbf{0}$. We are given that $v^\top M_Y v > 0$. As $v \in S$, $P_S v = v$. Therefore, $v^\top M_Y v = v^\top P_S M_Y P_S v > 0$. Let $A_S = P_S M_Y P_S$ restricted to $S$. $A_S$ is a positive semidefinite operator on $S$. The condition $v^\top A_S v > 0$ for $v \in S, v \neq 0$ implies that $A_S v \neq \mathbf{0}$ (because if $A_S v = \mathbf{0}$, then $v^\top A_S v = 0$, which contradicts $v^\top A_S v > 0$). Thus, $P_S M_Y P_S v \neq \mathbf{0}$, which means the condition $C_v$ is satisfied.

Since $v \in S$ and the condition $C_v$ for strict inequality is met, by Theorem 2, it follows that $v^\top \mathrm{Var}(\hat{\theta}_{X,Y})_{\theta_c,\theta_c} \, v < v^\top \mathrm{Var}(\hat{\theta}_X)_{\theta_c,\theta_c} \, v.$ $\square$

**Theorem 3.** *Define for any $m$, $I_X^{(m)} = \sum_{i=1}^m A_{c,i}^\top A_{c,i}$ and $I_Y^{(m)} = \sum_{j=1}^m B_{c,j}^\top B_{c,j}$. If $\mathrm{range}(I_Y^{(m)}) \not\subseteq \mathrm{range}(I_X^{(m)})$, then there exists a nonzero $v \in \mathbb{R}^{d_c}$ such that $v^\top I_Y^{(m)} v > v^\top I_X^{(m)} v.$*

*Proof.* Let $R_X := \mathrm{range}\big(I_X^{(m)}\big)$, $R_Y := \mathrm{range}\big(I_Y^{(m)}\big)$. By the assumption $R_Y \not\subseteq R_X$, choose a vector $w \in R_Y \setminus R_X$. Since $\mathbb{R}^{d_c}$ is a finite dimensional inner product space and $R_X$ is its finite dimensional subspace, we can decompose $w = w_{||} + v$ with $w_{||} \in R_X$ and $v \in R_X^{\perp}$. Because $w \notin R_X$, the orthogonal component $v$ is non-zero.

*(i) Term from $I_X^{(m)}$.* From the *Fundamental Theorem of Linear Algebra*, for any symmetric matrix $S$, $\ker S = \mathrm{range}(S)^{\perp}$; hence $R_X^{\perp} = \ker I_X^{(m)}$. Thus

$$v^{\top} I_X^{(m)} v = 0.$$

*(ii) Term from $I_Y^{(m)}$.* Because $w \in R_Y = \mathrm{range}(I_Y^{(m)})$, there exists $u$ with $w = I_Y^{(m)} u$. Suppose, for contradiction, that $I_Y^{(m)} v = 0$. Then $v \in \ker I_Y^{(m)} = R_Y^{\perp}$, so $v \perp w$. But $w \cdot v = (w_{||} + v) \cdot v = w_{||} \cdot v + \|v\|^2 = \|v\|^2 > 0$ because $v \perp w_{||}$ while $v \neq 0$. This contradicts $v \perp w$; therefore $I_Y^{(m)} v \neq 0$ and, by positive semidefiniteness,

$$v^{\top} I_Y^{(m)} v > 0.$$

Combining the above inequalities yields $v^{\top} I_Y^{(m)} v > v^{\top} I_X^{(m)} v$, with $v \neq 0$, which is the desired inequality. $\qquad\square$

# D  UML  Algorithm Pseudocode

In this section we present the full pseudocode for UML as shown in **??**.

---

**Algorithm 1** Pytorch Pseudocode for UML in the supervised setting

---

```
# f_img: image encoder (frozen or trainable)
# f_text: text encoder (frozen)
# is_trainable: True if f_img is trainable else False
# h: classification head

while not converged: # training loop
    x_img = fetch_next(image_loader) # image minibatch
    x_text = fetch_next(text_loader) # text minibatch (random/unaligned)

    z_img = f_img(x_img) # image embeddings
    z_text = f_text(x_text) # text embeddings

    logits_img = h(z_img) # predict image labels
    logits_text = h(z_text) # predict text labels

    loss_img = CE(logits_img, labels_img) # image classification loss
    loss_text = CE(logits_text, labels_text) # text classification loss
    loss = loss_img + lambda * loss_text # total loss

    loss.backward() # back-propagate
    update(h, f_img) if is_trainable else update(h) # SGD update

# Define Cross-Entropy loss
def CE(logits, labels):
    return -sum(labels * log_softmax(logits, dim=1)) / len(labels)
```

---

# E  Additional Experiments

## E.1  Improving Image Classification using Unpaired Texts (Unaligned encoders)

In this section we report image-classification results on ten benchmarks (see Appendix B.3), covering three settings:

1. **Full-dataset fine-tuning**: train both the vision backbone and classification head (Appendix E.1.1).

2. **Full-dataset linear probe**: train only the classification head (Appendix E.1.2).

3. **Few-shot linear probe**: train only the classification head under few-shot conditions (Appendix E.1.3).

In each setting, we compare UML with baselines across all datasets and multiple DINO-initialized vision backbones.

### E.1.1  Supervised Finetuning (across architectures)

In this section, we fine-tune both the vision backbone and the linear classifier on ten downstream tasks, comparing UML against strong image-only baselines. We evaluate four DINO-initialized backbones:

- ViT-B/16 in Table 5
- ViT-B/8 in  Table 6
- DINOv2 ViT-S/14 in Table 7
- DINOv2 ViT-B/14 in Table 8

Results for DINOv2 ViT-L/14 are omitted due to computational constraints. Across all backbones, UML consistently improves over the image-only baseline by leveraging unpaired text embeddings. For some backbones such as DINOv2 VIT-B/16, our head-initialization variant (*Ours (init)*) outperforms training using unpaired multimodal data from scratch (*Ours*), while in others it does not.

Table 5: **Full finetuning on classification with ViT-B/16 DINO and OpenLLaMA-3B**. We compare our proposed approach with the image-only baseline when fine-tuning on the target dataset. All vision encoders are initialized from DINO weights, and our approach leverages unpaired text data using OpenLLaMA-3B embeddings.

| Method | Stanford Cars | SUN397 | FGVC Aircraft | DTD | UCF101 | Food101 | Oxford Pets | Oxford Flowers | Caltech101 | Average |
|---|---|---|---|---|---|---|---|---|---|---|
| Unimodal | 78.41 | 63.99 | 62.12 | 74.17 | 81.43 | 82.38 | 92.00 | 98.24 | 96.31 | 81.01 |
| Ours | **82.56** | 67.04 | 67.38 | **76.42** | 84.06 | **81.79** | **93.20** | **98.98** | **97.04** | **83.16** |
| Ours (init) | 81.95 | **67.12** | **68.29** | 73.84 | **84.31** | 81.12 | 92.60 | 98.73 | 96.84 | 82.76 |

Table 6: **Full finetuning on classification with ViT-B/8 DINO and OpenLLaMA-3B**. We compare our proposed approach with the image-only baseline when fine-tuning on the target dataset. All vision encoders are initialized from DINO weights, and our approach leverages unpaired text data using OpenLLaMA-3B embeddings.

| | Dataset | | | | | | | | | |
|---|---|---|---|---|---|---|---|---|---|---|
| Method | Stanford Cars | SUN397 | FGVC Aircraft | DTD | UCF101 | Food101 | Oxford Pets | Oxford Flowers | Caltech101 | Average |
| Unimodal | 85.67 | 68.04 | 72.60 | 76.65 | 83.94 | **85.32** | 93.06 | 99.22 | 96.82 | 84.59 |
| Ours | **87.95** | **70.28** | 75.31 | **77.19** | 85.59 | 84.83 | 93.05 | **99.43** | 97.12 | 85.64 |
| Ours (init) | 87.44 | 70.03 | **76.09** | 76.24 | **86.49** | 84.71 | **93.81** | 99.27 | **97.16** | **85.69** |

Table 7: **Full finetuning on classification with ViT-S/14 DINOv2 and OpenLLaMA-3B**. We compare our proposed approach with the image-only baseline when fine-tuning on the target dataset. All vision encoders are initialized from DINOv2 weights, and our approach leverages unpaired text data using OpenLLaMA-3B embeddings.

| | Dataset | | | | | | | | | |
|---|---|---|---|---|---|---|---|---|---|---|
| Method | Stanford Cars | SUN397 | FGVC Aircraft | DTD | UCF101 | Food101 | Oxford Pets | Oxford Flowers | Caltech101 | Average |
| Unimodal | 79.45 | 66.20 | 66.99 | 72.16 | 83.18 | 80.65 | 90.67 | 99.18 | 95.45 | 81.54 |
| Ours | 84.87 | **66.72** | 71.54 | 74.14 | **84.77** | 81.16 | **91.87** | 99.55 | 97.03 | 83.52 |
| Ours (init) | **86.39** | 66.03 | **73.44** | **74.27** | 84.69 | **81.97** | 91.72 | **99.82** | **97.60** | **83.99** |

Table 8: **Full finetuning on classification with ViT-B/14 DINOv2 and OpenLLaMA-3B**. We compare our proposed approach with the image-only baseline when fine-tuning on the target dataset. All vision encoders are initialized from DINOv2 weights, and our approach leverages unpaired text data using OpenLLaMA-3B embeddings.

| | Dataset | | | | | | | | | |
|---|---|---|---|---|---|---|---|---|---|---|
| Method | Stanford Cars | SUN397 | FGVC Aircraft | DTD | UCF101 | Food101 | Oxford Pets | Oxford Flowers | Caltech101 | Average |
| Unimodal | 89.62 | **71.45** | 77.29 | 73.88 | 88.00 | 82.94 | 94.55 | **99.88** | 97.69 | 86.14 |
| Ours | **90.93** | 70.97 | 80.02 | 75.83 | 87.52 | **86.25** | **94.74** | **99.88** | 97.57 | **87.08** |
| Ours (init) | 90.73 | 70.92 | **80.23** | **75.87** | **87.60** | 83.43 | 94.47 | 99.80 | **97.93** | 86.77 |

### E.1.2 Linear Probing (across architectures)

In this section, we train only the linear classifier, on top of the frozen vision and language backbone, on ten downstream tasks, comparing UML against strong image-only baselines. We evaluate five DINO-initialized backbones:

- ViT-B/16 in Table 9

- ViT-B/8 in 
- DINOv2 ViT-S/14 in 
- DINOv2 ViT-B/14 in 
- DINOv2 ViT-L/14 in 

Across all backbones, UML consistently improves over the image-only baseline by leveraging unpaired text embeddings. For all backbones, our head-initialization variant (*Ours (init)*) outperforms training using unpaired multimodal data from scratch (*Ours*).

Table 9: **Full linear probing on classification with ViT-B/16 DINO and OpenLLaMA-3B**. We compare our proposed approach with the image-only baseline when training a linear probe on the target dataset. All vision encoders are initialized from DINO weights, and our approach leverages unpaired text data using OpenLLaMA-3B embeddings.

| Method | Stanford Cars | SUN397 | FGVC Aircraft | DTD | UCF101 | Food101 | Oxford Pets | Oxford Flowers | Caltech101 | Average |
|---|---|---|---|---|---|---|---|---|---|---|
| Unimodal | 67.10 | 64.63 | 56.02 | 72.42 | 81.27 | 74.96 | 93.07 | 98.32 | 95.01 | 78.08 |
| Ours | **68.71** | 65.14 | 57.42 | 72.95 | 82.06 | 75.30 | 93.18 | **98.46** | 96.19 | 78.82 |
| Ours (init) | 68.60 | **65.59** | **57.98** | **73.11** | **82.40** | **75.73** | **93.62** | 98.42 | **96.35** | **79.09** |

Table 10: **Full linear probing on classification with ViT-B/8 DINO and OpenLLaMA-3B**. We compare our proposed approach with the image-only baseline when training a linear probe on the target dataset. All vision encoders are initialized from DINO weights, and our approach leverages unpaired text data using OpenLLaMA-3B embeddings.

| Method | Stanford Cars | SUN397 | FGVC Aircraft | DTD | UCF101 | Food101 | Oxford Pets | Oxford Flowers | Caltech101 | Average |
|---|---|---|---|---|---|---|---|---|---|---|
| Unimodal | 72.01 | 67.19 | 62.02 | 76.18 | 82.95 | 78.57 | 91.99 | **98.78** | 96.23 | 80.66 |
| Ours | **72.93** | 68.17 | 63.49 | **77.13** | 83.16 | 79.87 | **92.59** | 98.50 | **96.47** | 81.37 |
| Ours (init) | 72.81 | **68.36** | **64.09** | 76.48 | **83.72** | **80.01** | 92.50 | 98.74 | 96.43 | **81.46** |

Table 11: **Full linear probing on classification with ViT-S/14 DINOv2 and OpenLLaMA-3B**. We compare our proposed approach with the image-only baseline when training a linear probe on the target dataset. All vision encoders are initialized from DINOv2 weights, and our approach leverages unpaired text data using OpenLLaMA-3B embeddings.

| | Dataset | | | | | | | | | |
|---|---|---|---|---|---|---|---|---|---|---|
| Method | Stanford Cars | SUN397 | FGVC Aircraft | DTD | UCF101 | Food101 | Oxford Pets | Oxford Flowers | Caltech101 | Average |
| Unimodal | 77.48 | 70.72 | 66.28 | 78.25 | 82.64 | 84.39 | 94.29 | 99.62 | 97.00 | 83.40 |
| Ours | 78.45 | 71.53 | 67.33 | 78.70 | 83.51 | 84.67 | 94.70 | 99.82 | 97.11 | 83.98 |
| Ours (init) | **78.58** | **72.24** | **67.50** | **79.51** | **83.57** | **84.74** | **94.78** | **99.89** | **97.15** | **84.22** |

Table 12: **Full linear probing on classification with ViT-B/14 DINOv2 and OpenLLaMA-3B**. We compare our proposed approach with the image-only baseline when training a linear probe on the target dataset. All vision encoders are initialized from DINOv2 weights, and our approach leverages unpaired text data using OpenLLaMA-3B embeddings.

| | Dataset | | | | | | | | | |
|---|---|---|---|---|---|---|---|---|---|---|
| Method | Stanford Cars | SUN397 | FGVC Aircraft | DTD | UCF101 | Food101 | Oxford Pets | Oxford Flowers | Caltech101 | Average |
| Unimodal | 85.46 | 75.42 | 72.34 | 79.73 | **87.26** | 88.70 | 95.56 | 99.76 | 97.81 | 86.89 |
| Ours | 85.40 | 75.22 | **75.22** | 80.73 | 87.21 | **89.02** | 95.83 | **99.88** | 97.85 | 87.37 |
| Ours (init) | **85.74** | **75.70** | 74.17 | **81.32** | **87.26** | 88.78 | **95.78** | **99.88** | **97.93** | **87.40** |

Table 13: **Full linear probing on classification with ViT-L/14 DINOv2 and OpenLLaMA-3B**. We compare our proposed approach with the image-only baseline when training a linear probe on the target dataset. All vision encoders are initialized from DINOv2 weights, and our approach leverages unpaired text data using OpenLLaMA-3B embeddings.

| | Dataset | | | | | | | | | |
|---|---|---|---|---|---|---|---|---|---|---|
| Method | Stanford Cars | SUN397 | FGVC Aircraft | DTD | UCF101 | Food101 | Oxford Pets | Oxford Flowers | Caltech101 | Average |
| Unimodal | 88.16 | 77.26 | 74.32 | 81.56 | 89.82 | 90.95 | 96.27 | 99.84 | 97.97 | 88.46 |
| Ours | **88.45** | 77.20 | 76.93 | 82.39 | **90.19** | 91.09 | **96.51** | **99.92** | **98.01** | 88.97 |
| Ours (init) | 87.99 | **77.75** | **77.20** | **82.51** | 90.17 | **91.29** | 96.32 | **99.92** | 97.93 | **89.01** |

### E.1.3 Few-shot Linear Probing (across architectures)

In this section, we train only the linear classifier, on top of the frozen vision and language backbone, for few-shot classification on ten downstream tasks, comparing UML against strong image-only baselines. We evaluate five DINO-initialized backbones: ViT-B/16 in Table 15, ViT-B/8 in Table 14, DINOv2 ViT-S/14 in Table 16, DINOv2 ViT-B/14 in Table 18, DINOv2 ViT-L/14 in Table 18. Across all backbones, UML consistently improves over the image-only baseline by leveraging unpaired text embeddings. For all backbones, our head-initialization variant (*Ours (init)*) outperforms training using unpaired multimodal data from scratch (*Ours*).

Table 14: **Linear evaluation of frozen features on 11 fine-grained benchmarks for few-shot learning.** We compare our proposed approach with the image-only baseline by training a linear classifier on top of frozen VIT-B/8 DINO features. Our method leverages unpaired text data using OpenLLaMA-3B

| Train Shot | Method | Stanford Cars | Sun397 | Fgvc Aircraft | Dtd | Ucf101 | Food101 | Imagenet | Oxford Pets | Oxford Flowers | Caltech101 | Average |
|---|---|---|---|---|---|---|---|---|---|---|---|---|
| 1 | Unimodal | 7.40 | 26.37 | 12.16 | 28.62 | 39.75 | 19.23 | 42.81 | 54.97 | 58.22 | 74.13 | 36.37 |
| | Ours | 7.71 | 28.01 | 13.56 | 33.22 | 42.08 | 21.13 | 43.27 | 55.85 | 58.61 | 77.51 | 38.10 |
| | Ours (init) | 9.24 | 34.23 | 14.49 | 36.27 | 47.55 | 24.81 | 46.75 | 60.09 | 61.59 | 80.23 | 41.52 |
| 2 | Unimodal | 14.43 | 37.96 | 20.28 | 39.80 | 53.03 | 30.62 | 54.75 | 68.12 | 77.59 | 81.91 | 47.85 |
| | Ours | 15.71 | 40.74 | 21.04 | 43.74 | 55.86 | 33.52 | 54.49 | 69.86 | 77.18 | 84.52 | 49.67 |
| | Ours (init) | 16.94 | 45.16 | 22.17 | 45.43 | 59.02 | 35.89 | 56.78 | 71.57 | 77.94 | 86.06 | 51.70 |
| 4 | Unimodal | 25.67 | 49.23 | 29.39 | 52.52 | 64.27 | 43.82 | 61.64 | 75.85 | 87.41 | 90.36 | 58.02 |
| | Ours | 27.30 | 51.23 | 31.43 | 54.31 | 66.72 | 45.58 | 61.51 | 77.51 | 87.96 | 91.36 | 59.49 |
| | Ours (init) | 28.54 | 53.68 | 31.31 | 56.13 | 67.47 | 47.40 | 62.84 | 79.10 | 88.29 | 91.98 | 60.67 |
| 8 | Unimodal | 41.04 | 56.86 | 40.03 | 61.15 | 72.39 | 54.47 | 66.10 | 82.30 | 93.95 | 92.28 | 66.06 |
| | Ours | 43.76 | 58.14 | 42.56 | 63.12 | 73.13 | 56.30 | 66.36 | 84.27 | 94.25 | 92.71 | 67.46 |
| | Ours (init) | 44.16 | 59.80 | 42.30 | 64.46 | 74.30 | 57.07 | 67.18 | 84.85 | 94.00 | 93.24 | 68.14 |
| 16 | Unimodal | 57.72 | 61.74 | 52.63 | 67.69 | 76.18 | 62.63 | 68.87 | 87.31 | 96.41 | 94.27 | 72.54 |
| | Ours | 60.11 | 63.21 | 54.53 | 69.33 | 78.13 | 63.74 | 69.44 | 87.73 | 96.89 | 94.54 | 73.76 |
| | Ours (init) | 60.36 | 64.26 | 54.81 | 70.27 | 78.76 | 64.13 | 70.05 | 88.23 | 96.63 | 94.73 | 74.22 |

Table 15: **Linear evaluation of frozen features on 11 fine-grained benchmarks for few-shot learning.** We compare our proposed approach with the image-only baseline by training a linear classifier on top of frozen VIT-B/16 DINO features. Our method leverages unpaired text data using OpenLLaMA-3B

| Train Shot | Method | Stanford Cars | Sun397 | Fgvc Aircraft | Dtd | Ucf101 | Food101 | Imagenet | Oxford Pets | Oxford Flowers | Caltech101 | Average |
|---|---|---|---|---|---|---|---|---|---|---|---|---|
| 1 | Unimodal | 6.28 | 22.43 | 9.72 | 29.22 | 37.85 | 15.40 | 38.67 | 60.12 | 54.62 | 73.25 | 34.76 |
| | Ours | 7.89 | 26.08 | 10.41 | 32.45 | 40.27 | 18.14 | 39.28 | 60.88 | 58.32 | 75.66 | 36.94 |
| | Ours (init) | 8.96 | 31.34 | 12.12 | 34.22 | 44.32 | 21.46 | 42.68 | 66.39 | 60.37 | 79.74 | 40.16 |
| 2 | Unimodal | 12.64 | 35.64 | 14.98 | 38.93 | 51.14 | 26.05 | 50.34 | 70.84 | 75.61 | 83.16 | 45.93 |
| | Ours | 14.38 | 38.62 | 17.00 | 40.37 | 54.28 | 29.24 | 50.83 | 72.88 | 77.14 | 85.95 | 48.07 |
| | Ours (init) | 15.99 | 42.31 | 17.65 | 42.89 | 56.46 | 32.15 | 52.90 | 74.82 | 77.32 | 87.34 | 49.98 |
| 4 | Unimodal | 22.60 | 45.95 | 24.27 | 50.30 | 63.00 | 38.51 | 57.99 | 80.14 | 85.60 | 89.67 | 55.80 |
| | Ours | 24.83 | 48.62 | 25.76 | 52.64 | 64.39 | 40.74 | 57.96 | 80.92 | 87.20 | 91.17 | 57.42 |
| | Ours (init) | 25.83 | 51.01 | 26.35 | 55.06 | 65.86 | 42.69 | 59.32 | 82.23 | 87.83 | 91.99 | 58.82 |
| 8 | Unimodal | 37.68 | 52.94 | 33.67 | 59.18 | 70.62 | 49.48 | 62.97 | 85.26 | 92.83 | 93.17 | 63.78 |
| | Ours | 39.31 | 55.31 | 35.56 | 60.48 | 71.88 | 50.46 | 63.08 | 86.25 | 93.23 | 93.47 | 64.90 |
| | Ours (init) | 40.50 | 57.03 | 35.64 | 62.27 | 73.18 | 51.50 | 64.09 | 86.93 | 93.59 | 93.71 | 65.84 |
| 16 | Unimodal | 52.48 | 58.27 | 45.34 | 64.81 | 75.72 | 56.24 | 66.36 | 88.57 | 95.90 | 94.27 | 69.80 |
| | Ours | 55.84 | 60.57 | 47.70 | 66.21 | 76.81 | 58.26 | 66.47 | 89.60 | 96.55 | 95.12 | 71.31 |
| | Ours (init) | 55.82 | 61.73 | 48.14 | 67.02 | 77.39 | 58.76 | 67.08 | 90.53 | 96.62 | 94.98 | 71.81 |

Table 16: **Linear evaluation of frozen features on 11 fine-grained benchmarks for few-shot learning.** We compare our proposed approach with the image-only baseline by training a linear classifier on top of frozen VIT-S/14 DINOv2 features. Our method leverages unpaired text data using OpenLLaMA-3B

| | | Dataset | | | | | | | | | | |
|---|---|---|---|---|---|---|---|---|---|---|---|---|
| Train Shot | Method | Stanford Cars | Sun397 | Fgvc Aircraft | Dtd | Ucf101 | Food101 | Imagenet | Oxford Pets | Oxford Flowers | Caltech101 | Average |
| 1 | Unimodal | 13.18 | 34.15 | 14.09 | 36.60 | 46.74 | 35.18 | 36.48 | 63.51 | 89.62 | 76.66 | 44.62 |
| | Ours | 14.95 | 37.25 | 14.88 | 38.93 | 49.18 | 37.91 | 38.35 | 68.92 | 91.42 | 84.04 | 47.58 |
| | Ours (init) | 16.49 | 41.79 | 15.63 | 42.04 | 52.33 | 42.27 | 42.69 | 73.59 | 93.64 | 84.52 | 50.50 |
| 2 | Unimodal | 24.68 | 47.88 | 23.09 | 47.75 | 56.81 | 48.54 | 50.41 | 75.32 | 96.02 | 86.90 | 55.73 |
| | Ours | 26.93 | 49.65 | 24.29 | 50.99 | 61.67 | 51.77 | 51.31 | 79.44 | 96.90 | 89.80 | 58.28 |
| | Ours (init) | 28.65 | 53.15 | 24.78 | 53.25 | 63.86 | 54.44 | 54.21 | 81.41 | 97.63 | 90.55 | 60.19 |
| 4 | Unimodal | 38.76 | 57.51 | 32.10 | 59.69 | 67.75 | 60.79 | 58.73 | 83.89 | 98.59 | 93.48 | 65.12 |
| | Ours | 41.69 | 58.87 | 33.38 | 61.58 | 69.60 | 62.69 | 59.69 | 86.27 | 98.84 | 94.56 | 66.71 |
| | Ours (init) | 43.17 | 60.89 | 33.86 | 62.43 | 71.13 | 63.88 | 61.38 | 87.36 | 99.17 | 94.96 | 67.82 |
| 8 | Unimodal | 54.56 | 63.00 | 45.05 | 64.78 | 74.19 | 68.06 | 64.53 | 88.68 | 99.27 | 94.35 | 71.65 |
| | Ours | 56.27 | 64.57 | 45.98 | 66.31 | 75.19 | 69.22 | 65.14 | 89.78 | 99.27 | 95.42 | 72.71 |
| | Ours (init) | 57.91 | 65.82 | 47.40 | 67.81 | 75.99 | 69.71 | 66.40 | 90.29 | 99.54 | 95.84 | 73.67 |
| 16 | Unimodal | 67.96 | 67.35 | 55.89 | 71.36 | 77.92 | 73.24 | 68.14 | 90.73 | 99.63 | 96.43 | 76.22 |
| | Ours | 69.42 | 68.50 | 58.54 | 72.24 | 78.69 | 73.80 | 68.70 | 91.87 | 99.72 | 96.63 | 77.80 |
| | Ours (init) | 70.32 | 69.19 | 58.74 | 73.17 | 79.58 | 74.51 | 69.44 | 92.47 | 99.82 | 96.80 | 78.81 |

Table 17: **Linear evaluation of frozen features on 10 fine-grained benchmarks for few-shot learning with DINOv2 ViT-B/14.** We compare our proposed approach with the image-only baseline by training a linear classifier on top of frozen VIT-B/14 DINOv2 features. Our method leverages unpaired text data using OpenLLaMA-3B

| | | Dataset | | | | | | | | | | |
|---|---|---|---|---|---|---|---|---|---|---|---|---|
| Train Shot | Method | Stanford Cars | Sun397 | Fgvc Aircraft | Dtd | Ucf101 | Food101 | Imagenet | Oxford Pets | Oxford Flowers | Caltech101 | Average |
| 1 | Unimodal | 22.42 | 43.03 | 15.79 | 38.85 | 58.57 | 48.71 | 52.26 | 76.47 | 97.12 | 83.64 | 53.69 |
| | Ours | 23.10 | 45.12 | 16.22 | 42.69 | 61.05 | 51.30 | 52.45 | 78.14 | 98.08 | 87.68 | 55.58 |
| | Ours (init) | 25.47 | 48.56 | 16.83 | 45.31 | 63.53 | 54.16 | 55.56 | 81.08 | 97.94 | 88.13 | 57.66 |
| 2 | Unimodal | 35.17 | 55.41 | 25.54 | 51.16 | 69.49 | 62.13 | 62.35 | 84.31 | 99.58 | 89.55 | 63.47 |
| | Ours | 37.38 | 56.98 | 25.88 | 54.65 | 70.61 | 63.89 | 63.21 | 85.50 | 99.70 | 92.02 | 64.98 |
| | Ours (init) | 38.78 | 59.81 | 26.00 | 55.61 | 71.38 | 66.54 | 65.06 | 86.49 | 99.62 | 92.79 | 66.21 |
| 4 | Unimodal | 51.40 | 63.68 | 34.25 | 61.25 | 76.32 | 71.60 | 68.86 | 89.05 | 99.76 | 94.51 | 71.07 |
| | Ours | 54.26 | 64.65 | 35.52 | 62.63 | 76.87 | 72.33 | 69.14 | 90.00 | 99.70 | 95.51 | 72.06 |
| | Ours (init) | 55.01 | 66.55 | 35.14 | 63.97 | 77.57 | 73.25 | 70.30 | 90.31 | 99.57 | 95.65 | 72.73 |
| 8 | Unimodal | 66.01 | 68.88 | 48.17 | 66.67 | 79.92 | 76.26 | 72.48 | 90.97 | 99.80 | 95.54 | 76.47 |
| | Ours | 68.53 | 69.75 | 50.88 | 68.46 | 81.44 | 77.34 | 73.12 | 92.39 | 99.70 | 96.20 | 77.78 |
| | Ours (init) | 67.91 | 70.66 | 51.26 | 69.56 | 81.85 | 77.95 | 73.75 | 92.50 | 99.68 | 96.51 | 78.16 |
| 16 | Unimodal | 77.31 | 72.17 | 62.38 | 73.76 | 83.80 | 80.74 | 75.15 | 93.34 | 99.81 | 97.40 | 81.59 |
| | Ours | 78.92 | 72.80 | 64.51 | 75.16 | 84.62 | 81.00 | 75.46 | 92.92 | 99.59 | 97.38 | 82.24 |
| | Ours (init) | 78.52 | 73.18 | 65.81 | 75.65 | 84.77 | 81.18 | 75.82 | 93.28 | 99.78 | 97.57 | 82.56 |

Table 18: **Linear evaluation of frozen features on 10 fine-grained benchmarks for few-shot learning with DINOv2 ViT-L/14.** We compare our proposed approach with the image-only baseline by training a linear classifier on top of frozen VIT-L/14 DINOv2 features. Our method leverages unpaired text data using OpenLLaMA-3B

| | | Dataset | | | | | | | | | | |
|---|---|---|---|---|---|---|---|---|---|---|---|---|
| Train Shot | Method | Stanford Cars | Sun397 | Fgvc Aircraft | Dtd | Ucf101 | Food101 | Imagenet | Oxford Pets | Oxford Flowers | Caltech101 | Average |
| 1 | Unimodal | 24.89 | 48.36 | 17.69 | 38.77 | 66.46 | 59.27 | 57.50 | 79.83 | 98.13 | 82.96 | 57.39 |
| | Ours | 25.88 | 49.63 | 18.08 | 42.93 | 69.18 | 60.12 | 58.37 | 83.51 | 98.42 | 86.23 | 59.24 |
| | Ours (init) | 27.90 | 52.86 | 18.95 | 43.18 | 70.98 | 63.17 | 60.80 | 83.86 | 98.59 | 88.17 | 60.85 |
| 2 | Unimodal | 39.95 | 58.95 | 26.87 | 50.18 | 75.79 | 70.74 | 67.14 | 84.71 | 99.74 | 89.82 | 66.39 |
| | Ours | 41.22 | 60.82 | 27.15 | 53.01 | 76.61 | 72.07 | 67.90 | 86.07 | 99.72 | 91.95 | 67.65 |
| | Ours (init) | 42.93 | 63.36 | 28.14 | 54.96 | 77.72 | 73.87 | 69.20 | 87.13 | 99.81 | 91.71 | 68.88 |
| 4 | Unimodal | 56.49 | 66.37 | 38.59 | 59.08 | 80.84 | 77.39 | 72.41 | 89.90 | 99.73 | 94.44 | 73.52 |
| | Ours | 58.19 | 67.36 | 39.57 | 61.78 | 81.36 | 78.19 | 72.82 | 90.99 | 99.76 | 95.27 | 74.53 |
| | Ours (init) | 58.60 | 68.84 | 39.19 | 62.77 | 81.50 | 78.99 | 73.63 | 90.74 | 99.88 | 96.02 | 75.02 |
| 8 | Unimodal | 70.00 | 70.71 | 51.57 | 66.47 | 83.84 | 81.69 | 76.02 | 93.53 | 99.89 | 95.55 | 78.93 |
| | Ours | 71.63 | 71.59 | 55.13 | 67.91 | 84.47 | 82.12 | 76.43 | 93.62 | 99.88 | 96.36 | 79.91 |
| | Ours (init) | 72.02 | 72.51 | 55.49 | 69.03 | 84.57 | 82.52 | 76.78 | 93.80 | 99.89 | 96.73 | 80.33 |
| 16 | Unimodal | 80.84 | 73.83 | 64.13 | 73.96 | 87.43 | 84.58 | 77.78 | 94.69 | 99.91 | 97.36 | 83.45 |
| | Ours | 81.85 | 74.39 | 69.45 | 74.70 | 87.35 | 84.58 | 78.35 | 94.59 | 99.89 | 97.61 | 84.28 |
| | Ours (init) | 82.76 | 74.80 | 69.42 | 74.88 | 87.65 | 84.96 | 78.58 | 94.42 | 99.81 | 97.62 | 84.49 |

## E.2 Improving Image Classification using Unpaired Texts (Aligned encoders)

### E.2.1 Supervised Finetuning

In this section, we fine-tune both the vision backbone and the linear classifier on nine downstream tasks, comparing UML against strong image-only baselines. We evaluate two different backbones: ResNet-50 and VIT-B/16.

As shown in Table 19, across all backbones, UML consistently improves over the image-only baseline by leveraging unpaired text embeddings. Further, our head-initialization variant (*Ours (init)*) outperforms training using unpaired multimodal data from scratch (*Ours*).

Table 19: **Supervised finetuning on 9 fine-grained classification benchmarks with CLIP.** We compare our proposed approach with the image-only baseline when fine-tuning on the target dataset. All vision encoders are initialized from CLIP ResNet50 weights, and our approach leverages unpaired text data using the corresponding CLIP text encoder.

| | Dataset | | | | | | | | | |
|---|---|---|---|---|---|---|---|---|---|---|
| Method | Stanford Cars | Sun397 | Fgvc Aircraft | Dtd | Ucf101 | Food101 | Oxford Pets | Oxford Flowers | Caltech101 | Average |
| Unimodal | 36.12 | 25.93 | 37.70 | 51.06 | 52.49 | 69.24 | 63.17 | 88.42 | 83.61 | 56.42 |
| Ours | 37.00 | 24.05 | 41.34 | 55.67 | 60.48 | 69.77 | 74.49 | 92.57 | 84.79 | 60.02 |
| Ours (init) | **72.75** | **62.33** | **66.58** | **56.50** | **67.54** | **76.95** | **86.97** | **94.80** | **87.95** | **74.71** |

Table 20: **Full linear probing on classification with CLIP ResNet-50 Image Encoder and Text encoder**. We compare our proposed approach with the image-only baseline when training a linear probe on the target dataset. All vision encoders are initialized from ResNet-50 weights, and our approach leverages unpaired text data using the corresponding CLIP text embeddings.

| Method | Stanford Cars | SUN397 | FGVC Aircraft | DTD | UCF101 | Food101 | Oxford Pets | Oxford Flowers | Caltech101 | Average |
|---|---|---|---|---|---|---|---|---|---|---|
| | | | | | | | | **Dataset** | | |
| Unimodal | 76.36 | 70.97 | 41.88 | 72.81 | 81.23 | 81.60 | 88.39 | **97.89** | 92.78 | 78.21 |
| Ours | 77.23 | 71.18 | 42.66 | 71.81 | 81.81 | 81.51 | 87.84 | 97.65 | 93.01 | 78.30 |
| Ours (init) | **79.14** | **73.83** | **42.81** | **73.76** | **82.13** | **82.44** | **90.90** | 97.69 | **94.19** | **79.65** |

### E.2.2 Linear Probing

In this section, we train only the linear classifier, on top of the frozen vision and language backbone from CLIP, on ten downstream tasks, comparing UML against strong image-only baselines. We evaluate two different backbones: ResNet-50 and VIT-B/16.

As shown in Table 20, across both backbones, UML consistently improves over the image-only baseline by leveraging unpaired text embeddings. Further, our head-initialization variant (*Ours (init)*) outperforms training using unpaired multimodal data from scratch (*Ours*).

### E.2.3 Few-shot linear Probing (across architectures)

In this section, we train only the linear classifier, on top of the frozen vision and language backbone from CLIP, for few-shot classification on ten downstream tasks, comparing UML against strong image-only baselines. We evaluate two different backbones: ResNet-50 and VIT-B/16.

As shown in Table 21 and Table 22, across both backbones, UML consistently improves over the image-only baseline by leveraging unpaired text embeddings. Further, our head-initialization variant (*Ours (init)*) outperforms training using unpaired multimodal data from scratch (*Ours*).

### E.3 Improving Visual Robustness Using Unpaired Texts

In this section, we evaluate the robustness of models trained with UML to test-time distribution shifts. We train a k-shot linear probe (where $k \in \{1, 2, 4, 8\}$) with DINOv2 on ImageNet and evaluate across four distribution-shifted target datasets: ImageNet-V2, ImageNet-Sketch, ImageNet-A, and ImageNet-R. Our method consistently improves robustness over the unimodal baseline (Figure 7, Figure 8, Figure 9 and Figure 10) across different training shots, indicating that language priors help capture more transferable features.

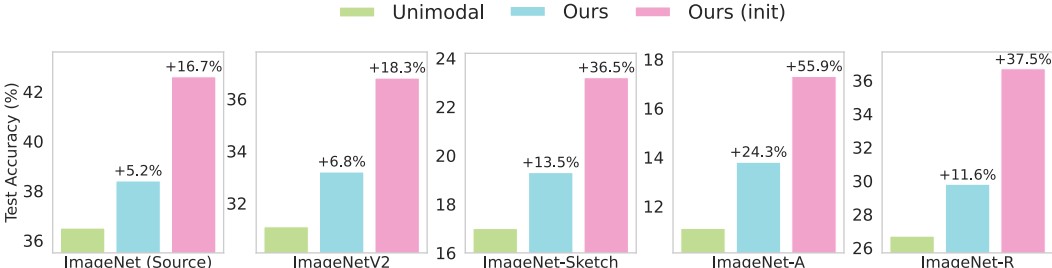

Figure 7: **Robustness under test-time distribution shifts.** Our approach (trained on 1-shot) is much more robust than its unimodal counterpart across four distribution-shuffled target test sets.

Table 21: **Linear evaluation of frozen features on 10 fine-grained benchmarks for few-shot learning.** We compare our proposed approach with the image-only baseline by training a linear classifier on top of frozen CLIP ResNet50 features. Our method leverages unpaired text data using the corresponding CLIP text encoder

| | | Dataset | | | | | | | | | | |
|---|---|---|---|---|---|---|---|---|---|---|---|---|
| Train Shot | Method | Stanford Cars | Sun397 | Fgvc Aircraft | Dtd | Ucf101 | Food101 | Oxford Pets | Imagenet | Oxford Flowers | Caltech101 | Average |
| 1 | Unimodal | 23.24 | 29.14 | 12.38 | 30.24 | 37.55 | 27.26 | 34.61 | 21.36 | 59.07 | 66.52 | 34.14 |
| | Ours | 36.32 | 45.40 | 16.84 | 40.92 | 53.19 | 49.76 | 53.03 | 36.48 | 68.56 | 76.80 | 47.73 |
| | Ours (init) | 57.88 | 64.59 | 22.23 | 50.85 | 65.99 | 76.73 | 86.59 | 60.92 | 81.08 | 83.79 | 65.06 |
| 2 | Unimodal | 38.37 | 43.83 | 18.63 | 40.33 | 53.25 | 44.60 | 47.75 | 32.62 | 75.03 | 78.90 | 47.33 |
| | Ours | 46.64 | 53.53 | 20.81 | 48.35 | 62.01 | 56.67 | 60.64 | 42.21 | 77.97 | 84.58 | 55.34 |
| | Ours (init) | 61.86 | 65.90 | 24.19 | 55.30 | 70.39 | 77.07 | 87.40 | 61.40 | 86.20 | 85.94 | 67.57 |
| 4 | Unimodal | 51.34 | 54.38 | 23.08 | 52.07 | 64.06 | 57.29 | 61.32 | 41.72 | 86.16 | 85.41 | 57.68 |
| | Ours | 55.21 | 59.48 | 24.77 | 56.78 | 67.65 | 62.68 | 67.31 | 47.04 | 86.46 | 87.23 | 61.46 |
| | Ours (init) | 65.80 | 68.11 | 27.49 | 60.13 | 73.62 | 77.79 | 86.54 | 62.37 | 91.60 | 87.57 | 70.10 |
| 8 | Unimodal | 61.74 | 61.47 | 30.22 | 60.15 | 70.16 | 64.63 | 68.94 | 49.48 | 92.20 | 89.14 | 64.81 |
| | Ours | 62.75 | 63.70 | 30.69 | 61.84 | 70.74 | 67.73 | 73.62 | 52.14 | 92.31 | 89.89 | 66.54 |
| | Ours (init) | 69.78 | 69.61 | 31.62 | 64.13 | 77.24 | 78.58 | 89.07 | 63.34 | 94.21 | 91.58 | 72.92 |
| 16 | Unimodal | 70.94 | 65.53 | 35.91 | 64.30 | 75.13 | 70.67 | 78.49 | 55.07 | 95.21 | 91.26 | 70.25 |
| | Ours | 71.58 | 67.08 | 36.23 | 65.62 | 76.09 | 71.63 | 79.52 | 56.92 | 95.44 | 91.94 | 71.20 |
| | Ours (init) | 74.56 | 71.33 | 37.13 | 68.09 | 78.66 | 79.06 | 89.71 | 64.31 | 96.17 | 93.31 | 75.23 |

Table 22: **Linear evaluation of frozen features on 10 fine-grained benchmarks for few-shot learning.** We compare our proposed approach with the image-only baseline by training a linear classifier on top of frozen CLIP VIT-B/16 features. Our method leverages unpaired text data using the corresponding CLIP text encoder

| | | Dataset | | | | | | | | | | |
|---|---|---|---|---|---|---|---|---|---|---|---|---|
| Train Shot | Method | Stanford Cars | Sun397 | Fgvc Aircraft | Dtd | Ucf101 | Food101 | Oxford Pets | Imagenet | Oxford Flowers | Caltech101 | Average |
| 1 | Unimodal | 31.53 | 33.51 | 17.76 | 31.72 | 43.64 | 39.40 | 37.43 | 27.65 | 67.95 | 71.68 | 40.23 |
| | Ours | 48.28 | 53.44 | 22.06 | 47.04 | 63.40 | 63.92 | 60.95 | 47.35 | 77.82 | 83.14 | 56.74 |
| | Ours (init) | 67.76 | 70.13 | 32.26 | 55.16 | 75.02 | 84.25 | 90.91 | 69.50 | 87.58 | 88.87 | 72.14 |
| 2 | Unimodal | 48.45 | 48.70 | 23.38 | 42.04 | 60.08 | 58.30 | 53.56 | 41.68 | 82.01 | 83.20 | 54.14 |
| | Ours | 57.89 | 59.95 | 27.19 | 52.27 | 69.60 | 71.18 | 66.78 | 54.24 | 87.43 | 90.20 | 63.67 |
| | Ours (init) | 70.75 | 71.52 | 33.99 | 60.17 | 78.37 | 85.39 | 90.67 | 70.19 | 92.18 | 90.09 | 74.33 |
| 4 | Unimodal | 61.64 | 60.66 | 31.01 | 54.37 | 70.49 | 71.91 | 69.35 | 52.15 | 90.99 | 91.08 | 65.36 |
| | Ours | 66.24 | 65.56 | 32.98 | 59.95 | 74.16 | 76.19 | 75.92 | 58.50 | 91.32 | 93.23 | 69.40 |
| | Ours (init) | 74.58 | 73.54 | 37.38 | 64.30 | 81.10 | 86.05 | 91.64 | 70.89 | 94.80 | 93.70 | 76.80 |
| 8 | Unimodal | 71.76 | 66.67 | 38.47 | 61.96 | 77.11 | 78.16 | 78.25 | 59.90 | 95.20 | 92.98 | 72.05 |
| | Ours | 72.77 | 69.50 | 39.09 | 64.89 | 79.01 | 80.07 | 80.85 | 62.63 | 94.98 | 94.36 | 73.82 |
| | Ours (init) | 78.43 | 75.07 | 41.77 | 68.50 | 83.41 | 86.87 | 92.55 | 71.97 | 96.94 | 95.27 | 79.08 |
| 16 | Unimodal | 78.76 | 71.49 | 44.74 | 68.79 | 80.43 | 82.08 | 85.16 | 63.87 | 96.97 | 94.54 | 76.68 |
| | Ours | 79.40 | 72.19 | 45.06 | 69.41 | 81.97 | 82.12 | 85.92 | 64.93 | 96.49 | 95.28 | 77.28 |
| | Ours (init) | 82.38 | 76.51 | 47.14 | 72.13 | 84.66 | 86.60 | 92.68 | 72.79 | 97.70 | 96.08 | 80.87 |

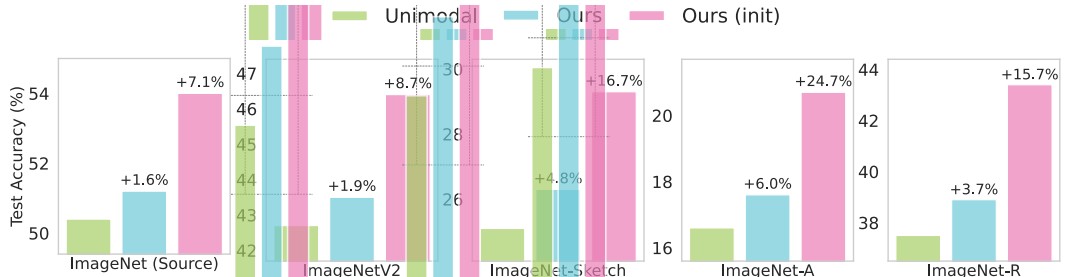

Figure 8: **Robustness under test-time distribution shifts.** Our approach (trained on 2-shots) is much more robust than its unimodal counterpart across four distribution-shuffled target test sets.

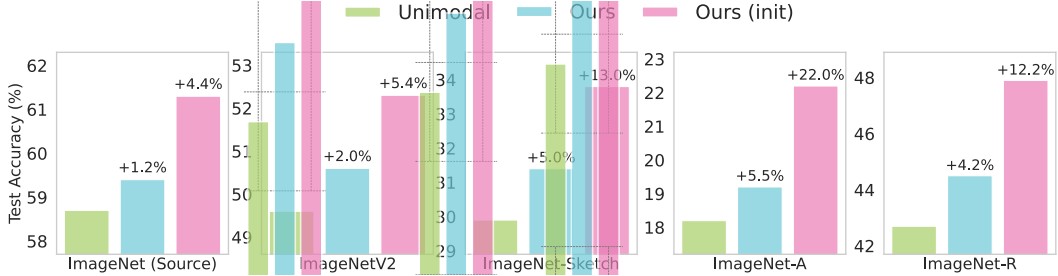

Figure 9: **Robustness under test-time distribution shifts.** Our approach (trained on 4-shots) is much more robust than its unimodal counterpart across four distribution-shuffled target test sets.

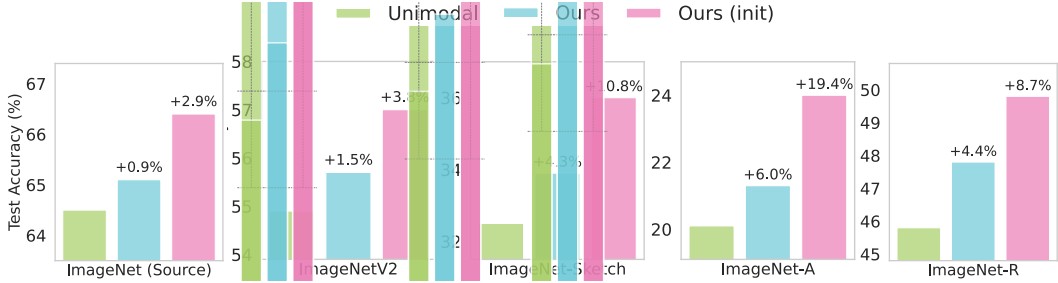

Figure 10: **Robustness under test-time distribution shifts.** Our approach (trained on 8-shots) is much more robust than its unimodal counterpart across four distribution-shuffled target test sets.

## E.4 Marginal Rate-of-Substitution Between Modalities

*How many words is an image worth?* In this section, we extend our results to evaluate image-text conversion ratios using test accuracy isolines on the remaining eight datasets. We measure these global equivalence ratios by fitting a plane to the accuracy values given the number of image and text shots. Figures 11 to 18 demonstrate the conversion ratios for DINOv2 VIT-S/14 as the vision backbone and OpenLLaMa-3B as the text backbone (unaligned encoders). Analogously, Figures 19 to 26 show the same ratios for CLIP ResNet-50 as the vision and text encoders (aligned encoders). As expected, with the fully aligned CLIP backbone, each image equates to far fewer text prompts than under the unaligned DINO setting, showing the higher efficiency of aligned embeddings.

### E.4.1 Unaligned Encoders

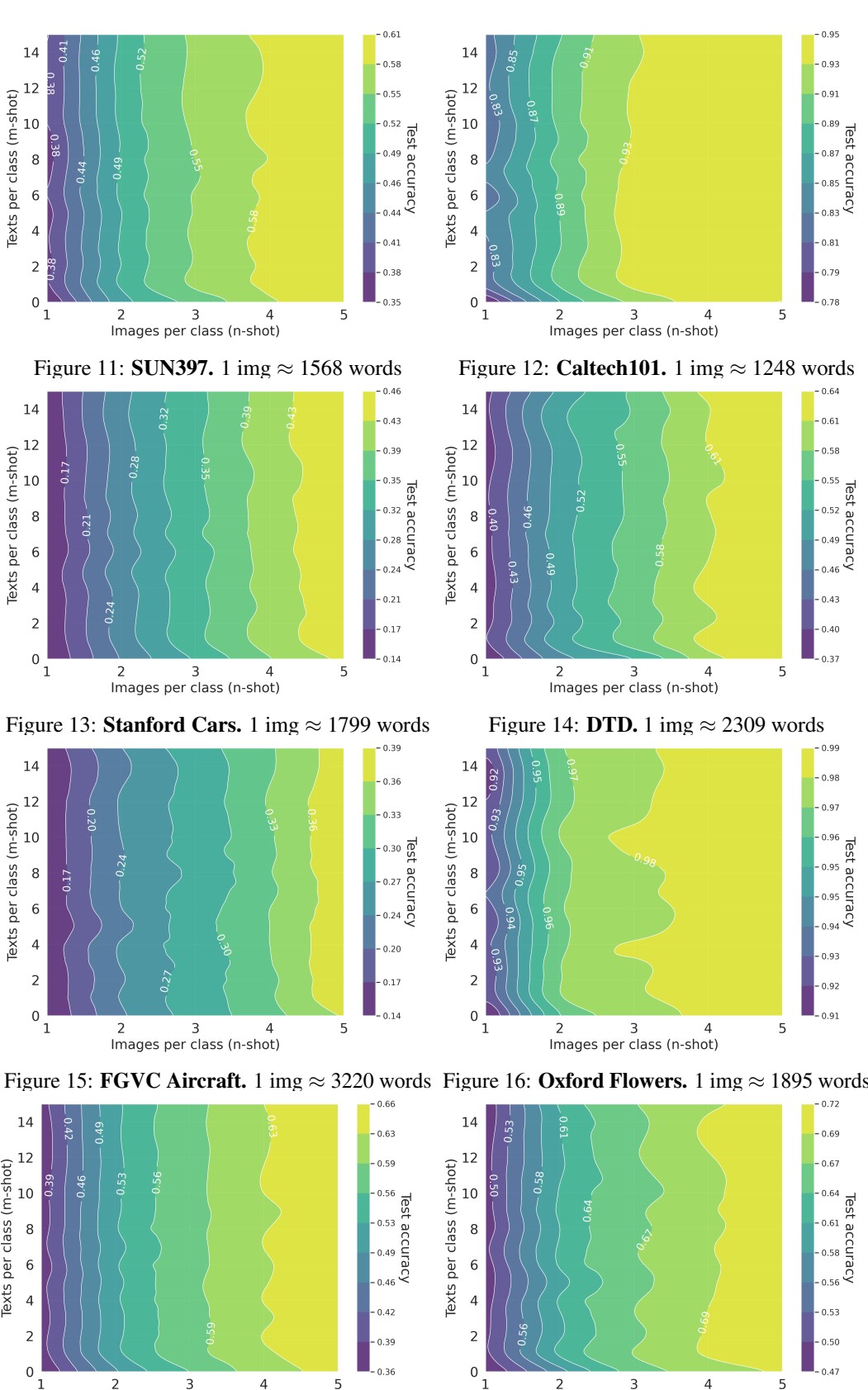

Figure 11: **SUN397.** 1 img ≈ 1568 words

Figure 12: **Caltech101.** 1 img ≈ 1248 words

Figure 13: **Stanford Cars.** 1 img ≈ 1799 words

Figure 14: **DTD.** 1 img ≈ 2309 words

Figure 15: **FGVC Aircraft.** 1 img ≈ 3220 words

Figure 16: **Oxford Flowers.** 1 img ≈ 1895 words

Figure 17: **Food101.** 1 img ≈ 2608 words

Figure 18: **UCF101.** 1 img ≈ 2617 words

### E.4.2 Aligned Encoders (CLIP)

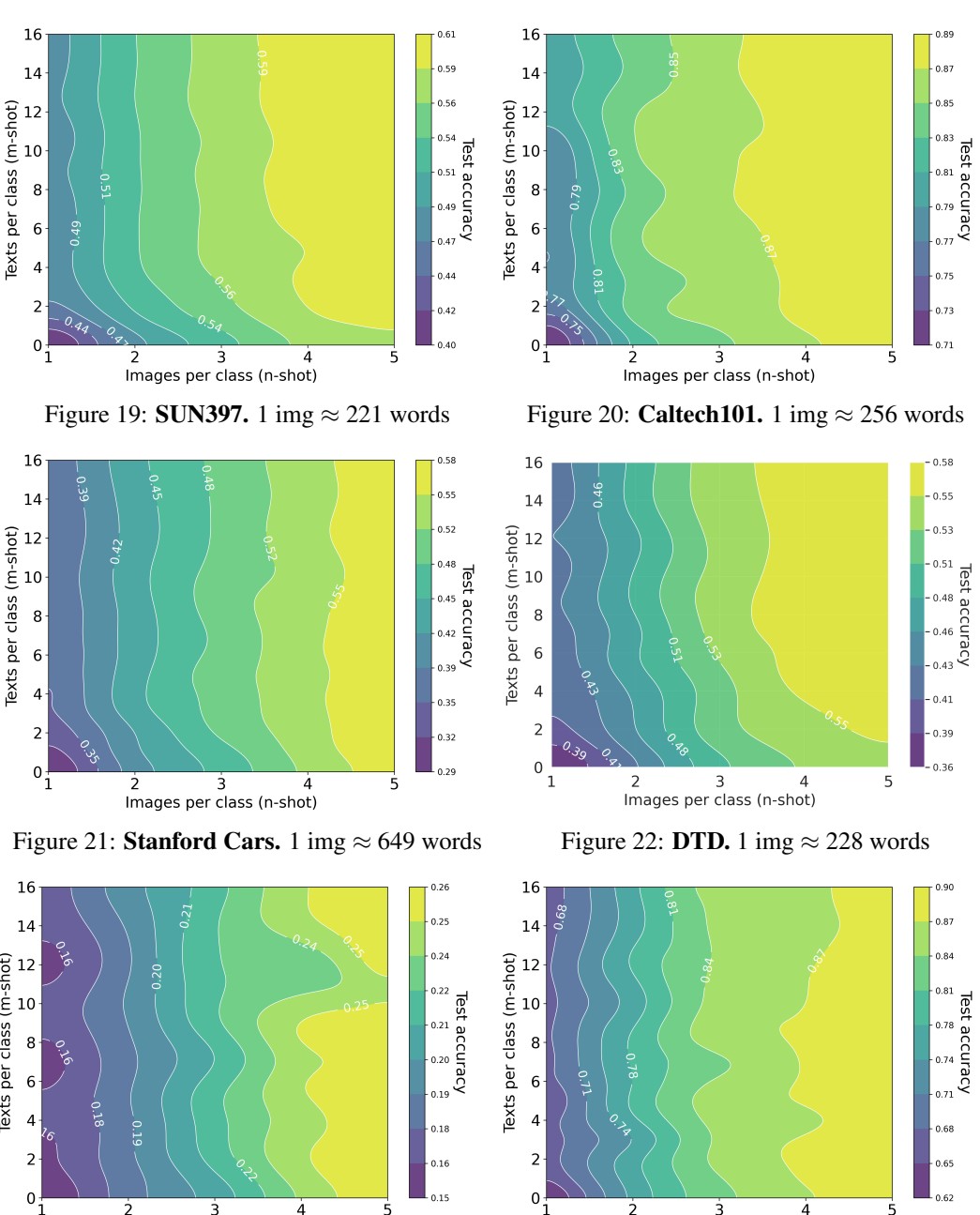

Figure 19: **SUN397.** 1 img ≈ 221 words

Figure 20: **Caltech101.** 1 img ≈ 256 words

Figure 21: **Stanford Cars.** 1 img ≈ 649 words

Figure 22: **DTD.** 1 img ≈ 228 words

Figure 23: **FGVC Aircraft.** 1 img ≈ 691 words

Figure 24: **Oxford Flowers.** 1 img ≈ 851 words

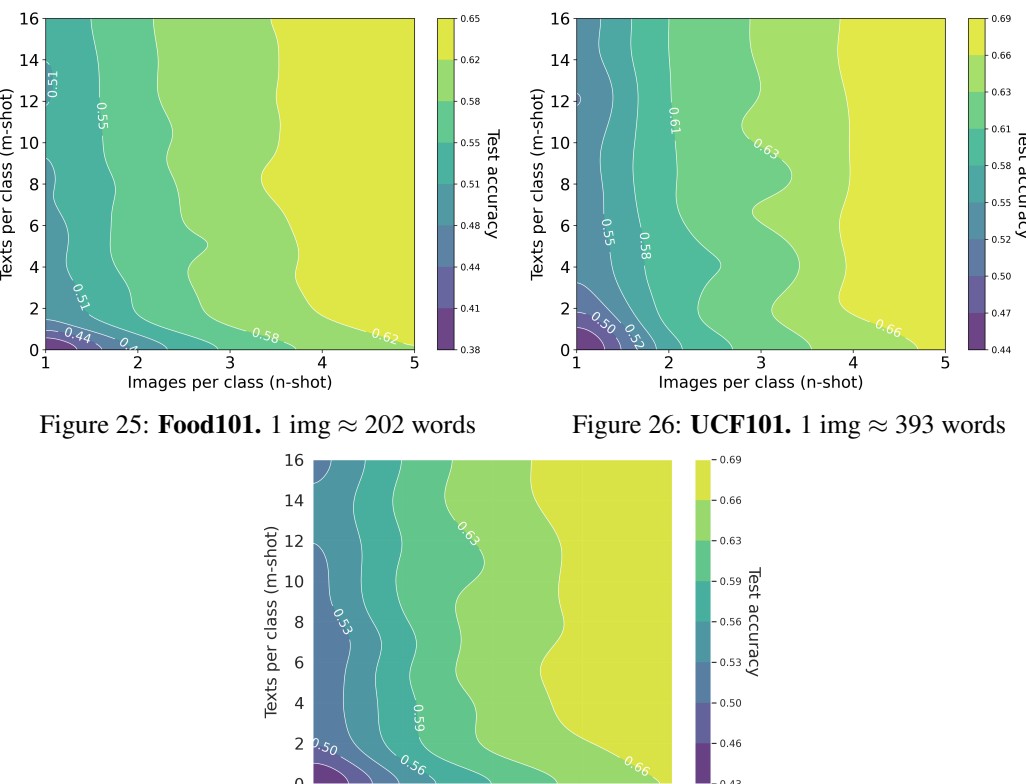

Figure 25: **Food101.** 1 img ≈ 202 words       Figure 26: **UCF101.** 1 img ≈ 393 words

Figure 27: **Oxford Pets.** 1 img ≈ 228 words

## E.5    Impact of Scaling Vision Backbone

In this section, we study how our method's performance scales with the size and architecture of the vision backbone. In addition to ViT-S/14 DINOv2, we extend our analysis to a range of ViT-based architectures, including ViT-B/14 and ViT-L/14 DINOv2 and ViT-B/16 and ViT-B/8 DINO models. To ensure a fair comparison, we follow the same training protocol as in previous experiments. Our method consistently outperforms the unimodal baselines in every setting. In few-shot linear probing across ViT-B/8, ViT-B/16, DINOv2-ViTs and ViT-L/14 backbones (Tables 14 to 18), we see clear gains. The same holds for full-dataset end-to-end fine-tuning of both encoder and head (Tables 5, 6 and 8 and **??**), and even when only the linear classifier is trained on the full splits (Tables 9 to 13).

## E.6    Impact of Varying Text Encoders

In this section, we study how our method's performance varies with different language models used for generating text embeddings. Through this experiment, we aim to understand how differences in embedding quality and model capacity affect the integration of textual information in our multimodal setup. Specifically, we cover LLMs with diverse architectures and scales, including BERT-Large, RoBERTa-Large and GPT-2 Large. As shown in Figure 28, adding unpaired text embeddings shows a significant boost in 1-shot accuracy and still decent gains at 16 shots on SUN397 dataset. Overall, OpenLLaMA-3B outperforms all other language models.

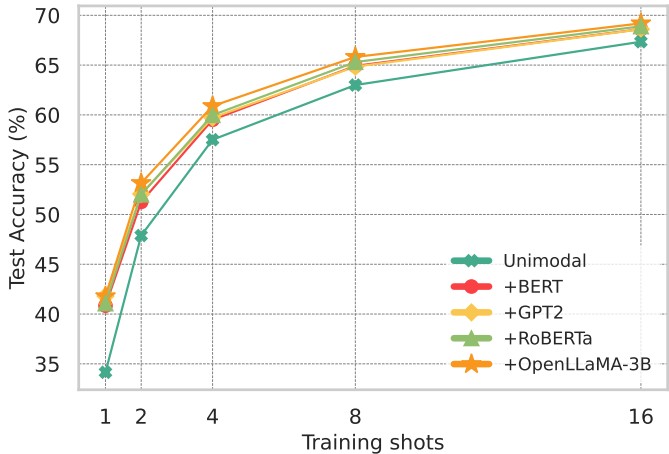

Figure 28: Few-shot classification accuracy on SUN397 using UML with unpaired, frozen embeddings from various pretrained language models.

## E.7   Learning with Coarse-Grained vs. Fine-Grained Textual Cues

Understanding the type of information extracted from textual cues is crucial to assessing the effectiveness of our multimodal approach. A key question is whether the model merely utilizes class names or goes beyond to capture richer, more descriptive features. To investigate this, we compare the performance of our method using two types of text templates: a vanilla template that consists solely of the class name (e.g., "a photo of a [class]") and descriptive templates generated from GPT-3, as detailed in Section ??. As shown in ?? and Figure 30, both multimodal approaches consistently outperform the unimodal baseline, with descriptions from GPT-3 offering a more substantial performance gain. This shows that leveraging richer, contextually diverse text cues can significantly enhance model performance, even in low-shot learning scenarios.

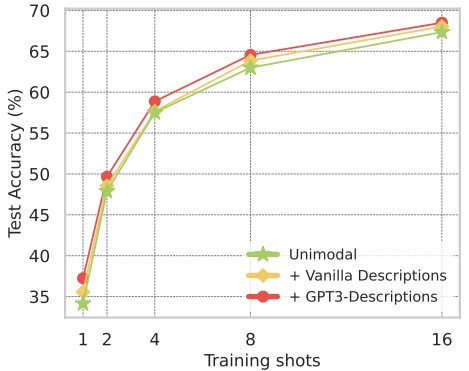
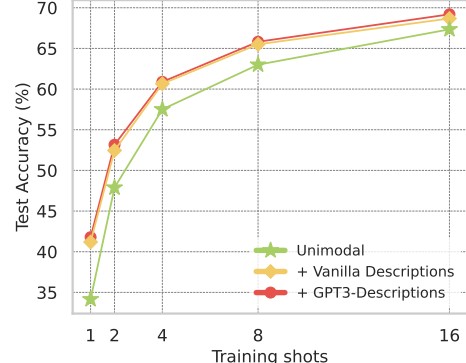

Figure 29: Few-shot SUN397 accuracy with UML using two levels of textual granularity: (a) vanilla class descriptions and (b) GPT-3–generated fine-grained descriptions.

Figure 30: Few-shot SUN397 accuracy with UML (init) using two levels of textual granularity: (a) vanilla class descriptions and (b) GPT-3–generated fine-grained descriptions.

## E.8   Impact on Performance with Increasing Unpaired Text Prompts

Here, we investigate how classification accuracy evolves as we augment each image with an increasing number of unpaired text prompts . Figure 31 shows these accuracy curves as we vary the number of

unpaired text prompts per image shot across five image-shot budgets. In every regime, our multimodal initialization ("Ours (init)") outperforms training the head from scratch, with most of the gain coming from the first few prompts and gains tapering off thereafter. Note that we do not enforce diversity or novelty in the unpaired text prompts—simply adding more sentences does not guarantee additional information.

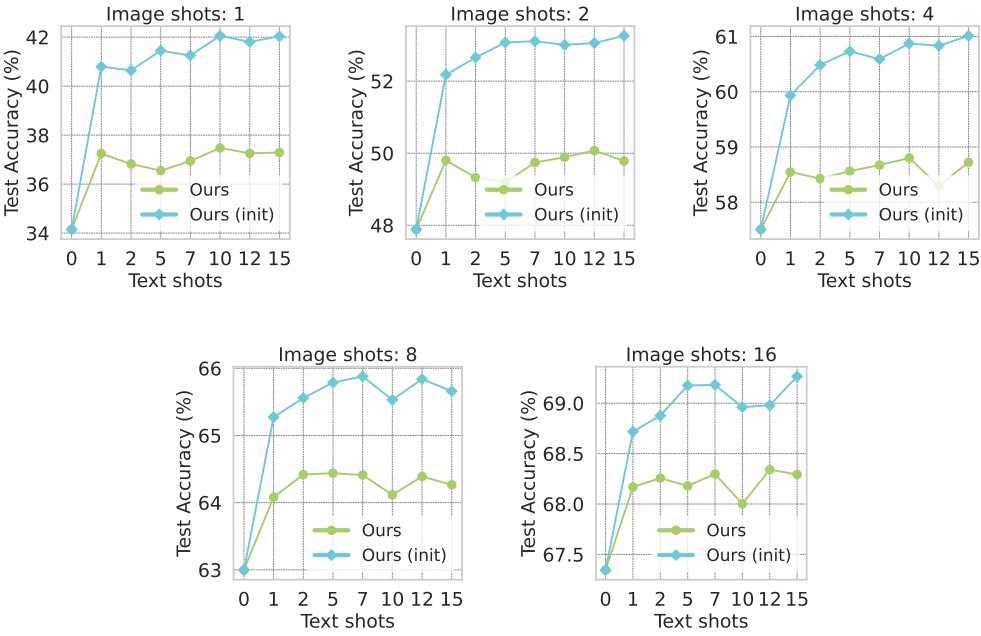

Figure 31: Classification accuracy as a function of the number of text prompts per image shot for the SUN397 Dataset.

## E.9 Additional Experiments for Audio-Visual Setting

In this section, we extend our unpaired multimodal framework to the tri-modal ImageNet–ESC benchmark, examining how unpaired audio and text signals can enhance image classification under both aligned (Appendix E.9.2) and unaligned encoders(Appendix E.9.1). We then reverse the setting, showing that unpaired visual and textual context likewise improves audio classification (Appendix E.9.3).

### E.9.1 Improving Image Classification with Unpaired Audio and Text (Unaligned encoders)

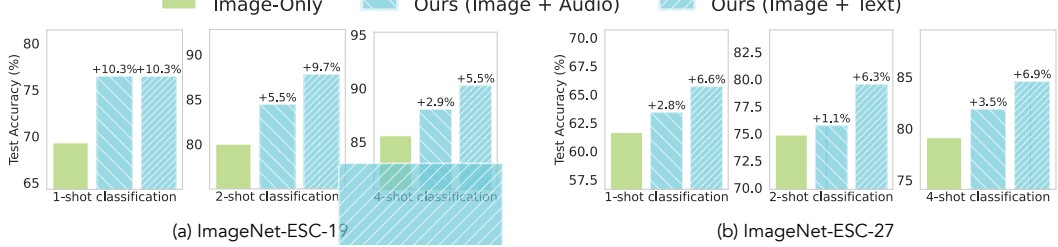

Figure 32: UML improves image classification using unpaired audio and text samples on both ImageNet-ESC-19 and ImageNet-ESC-27 benchmarks when trained on top of DINOv2 VIT-S/14 and OpenLLaMa-3B.

### E.9.2 Improving Image Classification with Unpaired Audio and Text (Aligned encoders)

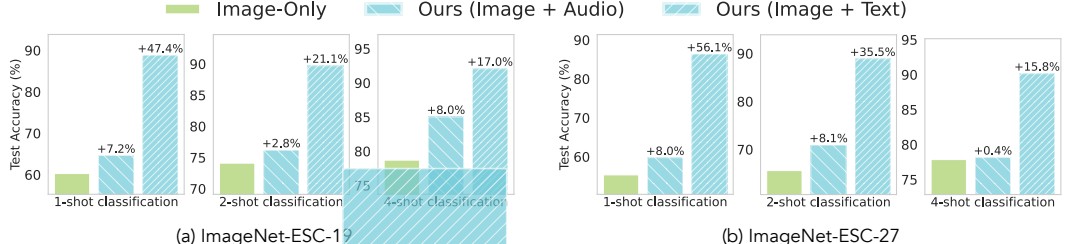

Figure 33: UML improves image classification using unpaired audio and text samples on both ImageNet-ESC-19 and ImageNet-ESC-27 benchmarks when trained on top of CLIP ResNet-50 image and text encoders

### E.9.3 Improving Audio Classification with Unpaired Image and Text (Aligned encoders)

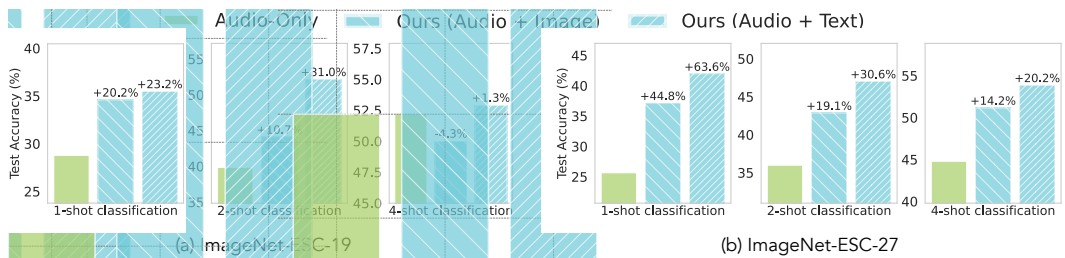

Figure 34: UML improves audio classification using unpaired image and text samples on both ImageNet-ESC-19 and ImageNet-ESC-27 benchmarks when trained on top of CLIP ResNet-50 image and text encoders

### E.10 Gaussian Experiments

Here, we shift our attention to a more nuanced and intriguing question: can incorporating unpaired multimodal data actually improve the *reconstruction* quality of a single modality? At first glance, this seems unlikely—why would adding data from a different modality make $X$ reconstruction better than training with $X$? Moreover, we push this question further: can incorporating data from a different modality, *while keeping the total dataset size fixed*, still improve the reconstruction of $X$ compared to using the same number of samples $X$ dataset alone? This setup isolates the importance of multimodal information from mere data scaling, and surprisingly, our experiments show that this improvement is indeed possible.

To investigate this, we design a synthetic experiment inspired by our theoretical framework in **??**. We generate data from two partially overlapping modalities, $X$ and $Y$, derived from a shared latent space $\theta_c$, while also containing unique components ($\theta_x$ and $\theta_y$). The observations follow the same linear structure as in our theory:

$$X_i = A_{c,i}\theta_c + A_{x,i}\theta_x + \epsilon_{X,i}$$
$$Y_j = B_{c,j}\theta_c + B_{y,j}\theta_y + \epsilon_{Y,j}$$

The overlap ratio, denoted as $p$, controls how much of the shared latent dimensions are jointly captured by both $X$ and $Y$. We set $p = 0.2$, meaning that only 20% of the shared latent dimensions are observed by both modalities, while the remaining 40% are exclusively captured by $X$ and $Y$ respectively. This structured overlap ensures that neither modality alone can fully reconstruct the shared latent space, forcing the model to integrate complementary information from both.

Our architecture consists of a shared autoencoder with separate input projections for $X$ and $Y$. Each modality is first encoded through a modality-specific linear projection layer, followed by a shared latent encoder composed of two layers with ReLU nonlinearity. The encoded representation is then passed through a decoder, also consisting of two linear layers, to reconstruct the input. We use separate heads for the final reconstruction, while keeping the latent space shared to promote cross-modal alignment.

As shown in Figure 35, the surprising outcome is that training on both modalities, even when they are unpaired, consistently improves the reconstruction of $X$ compared to training solely on $X$. More strikingly, this improvement holds even when the total number of training samples is fixed, with half the data coming from $X$ and half from $Y$; showing that the model is not just benefiting from increased data quantity but from the diversity and complementary information provided by the second modality.

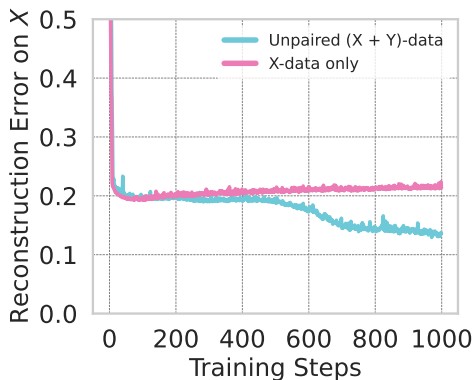

Figure 35: Training on $N/2$ samples from $X$ and $N/2$ unpaired samples from $Y$ improves test reconstruction on $X$, more than training on $N$ samples from $X$.

# F   Analysis of the Learned Classifier

## F.1   Change in Decision Boundaries with Unpaired Data from Another Modality

Our decision boundary visualizations are constructed by projecting the high-dimensional embedding space of a given classifier to a 2D plane. Axis 1 is computed as the normalized difference between the classifier weights of the two selected classes, representing the primary decision direction. Axis 2 is chosen to be orthogonal to Axis 1, constructed from the difference between the class mean embeddings after removing the component parallel to Axis 1. This orthogonalization ensures that the two axes capture complementary aspects: Axis 1 reflects the primary model decision boundary, while Axis 2 captures the variation orthogonal to that decision. The final 2D projection matrix combines these two vectors as columns, and embedding vectors are then mapped to this plane using a simple dot product. Figure 36 and Figure 37 show the change in decision boundary when adding unpaired textual information for 2-shot classification on top of frozen CLIP ResNet-50 features for DTD and Oxford Flowers datasets.

## F.2   What do models learn from unpaired data?

To understand what the model is truly learning and how its weights evolve, we develop and analyze three key metrics: functional margin, silhouette score, and class-prototype vectors. These metrics inform on how well the model distinguishes between classes and how text information influences the structure of feature-space

**Functional margin.** This quantifies how confidently a model separates a given sample from the decision boundary. For a sample $i$ belonging to class $y$, we calculate the margin relative to the next highest competing class. Specifically, we identify the second-highest logit among the incorrect classes, denoted as class $j^*$, and compute the functional margin as

$$\gamma_i = \frac{w_y^T x_i - w_{j^*}^T x_i}{\|w_y - w_{j^*}\|_2} \tag{3}$$

where $w_y^T x_i$ represents the logit for the true class, while $w_{j^*}^T x_i$ represents the highest logit among the competing classes. Larger margins indicate more confident and robust classification, while smaller

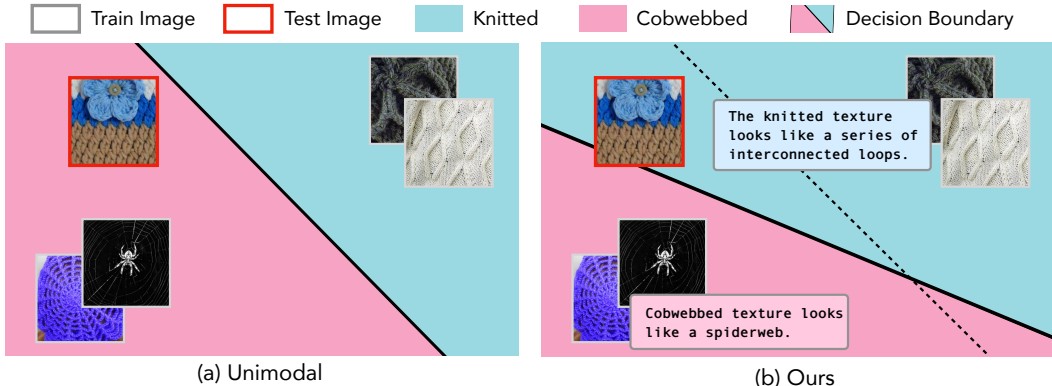

Figure 36: **Impact of unpaired text on decision boundaries (CLIP ResNet50).** (Left) Visual features alone learn ambiguous class boundaries between knitted and cobwebbed. (Right) Adding unpaired text sharpens the boundary, leveraging semantic cues to better distinguish similar categories

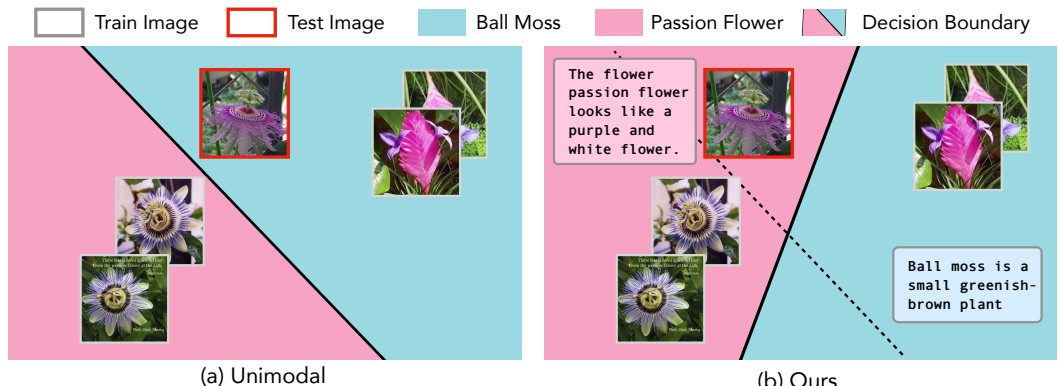

Figure 37: **Impact of unpaired text on decision boundaries (CLIP ResNet50).** (Left) Visual features alone learn ambiguous class boundaries between ball moss and passion flower. (Right) Adding unpaired text sharpens the boundary, leveraging semantic cues to better distinguish similar categories

margins imply that the sample lies closer to a misclassification boundary. As shown in Figure 38, both *Ours* and *Ours (init)* exhibit substantially larger classification margins than the unimodal baseline, demonstrating that augmenting primary-modality training with unpaired multimodal data improves confidence in predictions over the primary modality.

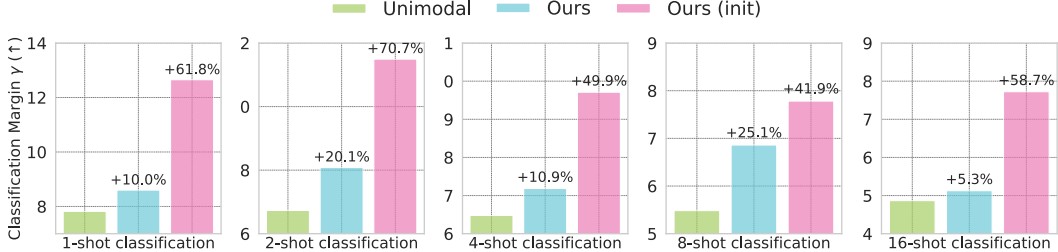

Figure 38: Functional margin of the linear head trained on SUN397 dataset for few-shot classification significantly increases when training with both UML and UML with linear head initialization.

**Silhouette Score and DB-Index.** The Silhouette Score indicates how well-separated the clusters are, while the DB-Index measures intra-class compactness versus inter-class separation. Higher silhouette and lower DB-Index values mean better-defined clusters, indicating that text helps tighten intra-class spread and widen inter-class gaps. As shown in Figure 39 and Figure 40, both *Ours*

and *Ours (init)* exhibit reduced intra-class distances and increased inter-class separations, further confirming improved class separability.

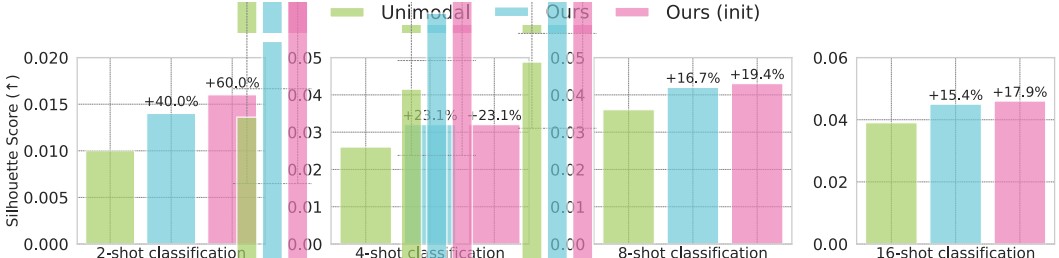

Figure 39: Silhouette Score of the linear head trained on SUN397 dataset for few-shot classification significantly increases when training with both UML and UML with linear head initialization.

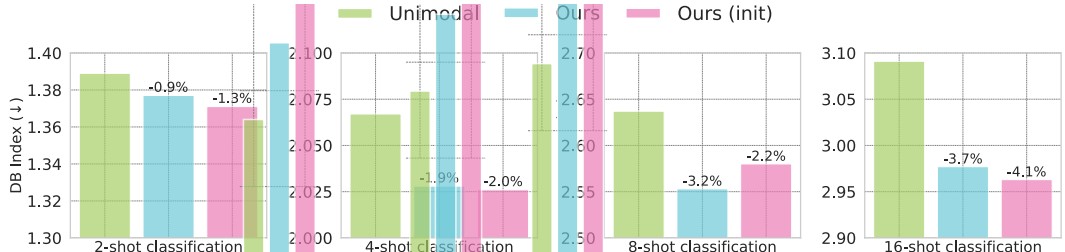

Figure 40: DB-Index of the linear head trained on SUN397 dataset for few-shot classification significantly improves when training with both UML and UML with linear head initialization.

**Class-Prototype Vectors.** These vectors are the rows of the final linear layer's weight matrix, representing the class centroids in the shared embedding space. We compute a heatmap of inner products between class prototypes and average text embeddings of the corresponding class to assess how well text features align with class centers. This helps reveal how the model organizes multimodal information. Figure 41 shows a pronounced diagonal structure, indicating that each class's text embedding aligns closely with the learned weights of the model.

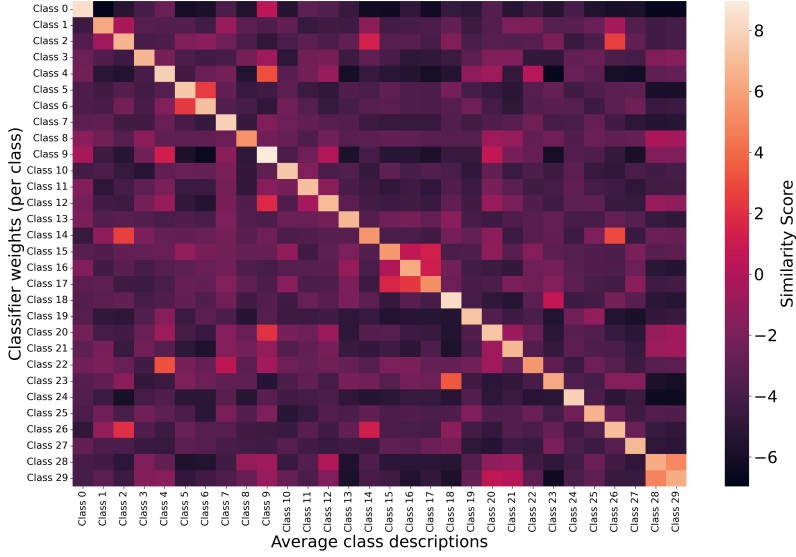

Figure 41: Inner products between each linear-head weight vector and its class's mean text embedding, demonstrating that text features align well with class prototypes.

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
