# OpenReview forum: "Better Together: Leveraging Unpaired Multimodal Data for Stronger Unimodal Models"
_NeurIPS.cc/2025/Workshop/UniReps — UniReps2025_

### Official Review · Reviewer_hBxy · 2025-09-16
**Paper offers a way to leverage unpaired data to improve SSL representations across modalities**

**Confidence:** 4

**Review:**

Strengths:
- paper is well-written and clear, overall presentation is good
- topic is well-aligned with the interest of this community and the context is well-presented
- proposed setup is simple, intuitive and results show a significant impact on downstream performance.
- nice to see results on text, images and audio modalities

Weaknesses:
- the second (theoretical) contribution is underrepresented in this current short submission
- related work [28] in the appendix seems very related, and should be compared against the proposed method
- conceptually the method is simple but requires some text template engineering

Minor:
- in the introduction authors report results on 10 & 11 datasets which is not aligned with the empirical setup showed in section 4
- would be curious to know why authors think the _Ours (init)_ method surpasses the _Ours_?

**Score:**

4

**Topic Fit:**

3

---

### Official Review · Reviewer_8pVK · 2025-09-16
**Review of Submission31**

**Confidence:** 4

**Review:**

**Summary:**

The paper introduces Unpaired Multimodal Learner (UML), a training framework for enhancing unimodal models by leveraging the structure coming from unpaired multimodal data. UML is implemented with a shared-weights multimodal network, trained with supervision coming from separate modalities. The authors provide theoretical results backing their method under simplified assumptions, and empirically validate it on different tasks and modalities, including vision, text and audio.

**Strengths:**

- The paper is clearly written and easy to follow, with helpful figures and detailed appendices that support the main text;

- It proposes an intriguing novel idea: using unpaired multimodal data to improve unimodal models, moving beyond the standard reliance on paired datasets;

- The contributions are supported by both solid theoretical analysis and extensive experiments across multiple modalities, including vision, text, and audio;

- The work addresses a highly practical problem, showing how abundant unpaired data can be leveraged when paired data is scarce or expensive to obtain.

**Weaknesses:**

- The paper does not clearly describe the UML architecture itself. Beyond mentioning pretrained encoders and a shared head, it is left vague whether UML involves a specific architectural choice (e.g., transformer, MLP) or is purely a training paradigm. This lack of detail makes it difficult to assess reproducibility and architectural contributions;

- The section on transfer learning from BERT to ViT, while interesting, feels somewhat disconnected from the main proposal, as it does not actually involve the UML framework and reads more like an adjacent experiment.

**Score:**

5

**Topic Fit:**

3

---

### Official Review · Reviewer_yfmN · 2025-09-17

**Confidence:** 4

**Review:**

Paper summary:
In this paper, the authors present UML, a framework for using "unpaired" data from another modality to improve performance on unimodal classification tasks. The authors first theoretically justifies that, under linear data generation assumption, the proposed method will help, and then conducted extensive experiments over many vision tasks and audio tasks and over many pre-trained model combinations to show that UML consistently improved unimodal performance. The paper further included additional analysis on UML's improved robustness, transfer learning, and marginal image-to-word substitution rate.

Strengths:
1. The proposed method is extensively tested on over 10 commonly used image classification datasets, and it consistently outperforms baselines across all of them
2. In addition to just visual-language, the paper also included experiments that shows that visual and language information can help train audio models.
3. There is theoretical proof that the proposed method improves performance in the Appendix.
4. There are extensive results and documentation on experiments spanning across many combinations of visual model, language model, and method of fine-tuning (full fine-tune vs linear layer only) in the Appendix. Further analysis was conducted on transfer learning and robustness, where the proposed method also performs well.

Weaknesses:
1. The term "unpaired" is very misleading in this paper. While I believe that the proposed method has merits in improving model performance, I would not call it a method that can work on "unpaired" multimodal data. Although the additional data from another modality (e.g. text) does not directly pair with each image, it seems like data from both modalities has to be labeled in a shared set of classes. Therefore, technically, each image can be viewed as "paired" with text data with the same label. In practice, as mentioned in the Appendix, the text used for training is generated by LLM given the set of labels to provide useful information about the classes. Thus, I also believe that the term "unaligned multimodal signals" used in the abstract is misleading, as the multimodal signals in this paper's settings are clearly aligned through shared labels.
2. The applicability of the method is limited to classification tasks with labeled data. In practice, labeled images are often much scarcer than images with paired captions (and you can always convert the former into latter)
3. Latex reference error on line 794 in Appendix

**Score:**

3

**Topic Fit:**

2

---

### Official Review · Reviewer_J5QH · 2025-09-19
**This paper proposes Unpaired Multimodal Representation Learning (UML), which uses unpaired data from other modalities (like text or audio) to improve unimodal models.**

**Confidence:** 4

**Review:**

This paper proposes Unpaired Multimodal Representation Learning (UML), which uses unpaired data from other modalities (like text or audio) to improve unimodal models. The method is simple — different modalities share the same classifier head and are trained alternately. The idea is backed up with some theory and tested on multiple benchmarks, showing consistent gains and better robustness. I especially liked the analysis of “exchange rates” between modalities (e.g., how many words equal an image) and the transfer experiment from BERT to ViT.

**Score:**

4

**Topic Fit:**

2